# Neural Architecture Dilation for Adversarial Robustness

**Yanxi Li** [1], **Zhaohui Yang** [2,3], **Yunhe Wang** [2], **Chang Xu** [1]

[1] School of Computer Science, University of Sydney, Australia
[2] Huawei Noah's Ark Lab
[3] Key Lab of Machine Perception (MOE), Department of Machine Intelligence,
Peking University, China
`yali0722@uni.sydney.edu.au, zhaohuiyang@pku.edu.cn,`
`yunhe.wang@huawei.com, c.xu@sydney.edu.au`

## Abstract

With the tremendous advances in the architecture and scale of convolutional neural networks (CNNs) over the past few decades, they can easily reach or even exceed the performance of humans in certain tasks. However, a recently discovered shortcoming of CNNs is that they are vulnerable to adversarial attacks. Although the adversarial robustness of CNNs can be improved by adversarial training, there is a trade-off between standard accuracy and adversarial robustness. From the neural architecture perspective, this paper aims to improve the adversarial robustness of the backbone CNNs that have a satisfactory accuracy. Under a minimal computational overhead, the introduction of a dilation architecture is expected to be friendly with the standard performance of the backbone CNN while pursuing adversarial robustness. Theoretical analyses on the standard and adversarial error bounds naturally motivate the proposed neural architecture dilation algorithm. Experimental results on real-world datasets and benchmark neural networks demonstrate the effectiveness of the proposed algorithm to balance the accuracy and adversarial robustness.

## 1 Introduction

In the past few decades, novel architecture design and network scale expansion have achieved significant success in the development of convolutional neural networks (CNN) [12, 13, 11, 23, 9, 31, 1, 35, 17, 15]. These advanced neural networks can already reach or even exceed the performance of humans in certain tasks [10, 21]. Despite the success of CNNs, a recently discovered shortcoming of them is that they are vulnerable to adversarial attacks. The ingeniously designed small perturbations when applied to images could mislead the networks to predict incorrect labels of the input [7]. This vulnerability notably reduces the reliability of CNNs in practical applications. Hence developing solutions to increase the adversarial robustness of CNNs against adversarial attacks has attracted particular attention from the researchers.

Adversarial training can be the most standard defense approach, which augments the training data with adversarial examples. These adversarial examples are often generated by fast gradient sign method (FGSM) [7] or projected gradient descent (PGD) [18]. Tramèr et al. [24] investigates the adversarial examples produced by a number of pre-trained models and developed an ensemble adversarial training. Focusing on the worst-case loss over a convex outer region, Wong and Kolter [27] introduces a provable robust model. There are more improvements of PGD adversarial training techniques, including Lipschitz regularization [6] and curriculum adversarial training [2]. In a recent study by Tsipras et al. [25], there exists a trade-off between standard accuracy and adversarial robustness. After the networks have been trained to defend against adversarial attacks, their performance over natural image classification could be negatively influenced. TRADES [32] theoretically studies this trade-off by introducing a boundary error between the natural (i.e. standard) error and the robust error. Instead of directly adjusting the trade-off, the friendly adversarial training (FAT) [33] proposes to exploit weak adversarial examples for a slight standard accuracy drop.

35th Conference on Neural Information Processing Systems (NeurIPS 2021).

Numerous efforts have been made to defend the adversarial attacks by carefully designing various training objective functions of the networks. But less noticed is that the neural architecture actually bounds the performance of the network. Recently there are a few attempts to analyze the adversarial robustness of the neural network from the architecture perspective. For example, RACL [5] applies Lipschitz constraint on architecture parameters in one-shot NAS to reduce the Lipschitz constant and improve the robustness. RobNet [8] search for adversarially robust network architectures directly with adversarial training. Despite these studies, a deeper understanding of the accuracy and robustness trade-off from the architecture perspective is still largely missing.

In this paper, we focus on designing neural networks sufficient for both standard and adversarial classification from the architecture perspective. We propose **n**eural **a**rchitecture **d**ilation for **a**dversarial **r**obustness (NADAR). Beginning with the backbone network of a satisfactory accuracy over the natural data, we search for a dilation architecture to pursue a maximal robustness gain while preserving a minimal accuracy drop. Besides, we also apply a FLOPs-aware approach to optimize the architecture, which can prevent the architecture from increasing the computation cost of the network too much. We theoretically analyze our dilation framework and prove that our constrained optimization objectives can effectively achieve our motivations. Experimental results on benchmark datasets demonstrate the significance of studying the adversarial robustness from the architecture perspective and the effectiveness of the proposed algorithm.

## 2 Related Works

### 2.1 Adversarial Training

FGSM [7] claims that the adversarial vulnerability of neural networks is related to their linear nature instead of the nonlinearity and overfitting previously thought. A method to generate adversarial examples for adversarial training is proposed based on such a perspective to reduce the adversarial error. PGD [18] studies the adversarial robustness from the view of robust optimization. A first-order gradient-based method for iterative adversarial is proposed. FreeAT [22] reduces the computational overhead of generating adversarial examples. The gradient information in network training is recycled to generate adversarial training. With this gradient reusing, it achieves 7 to 30 times of speedup.

However, the adversarial robustness comes at a price. Tsipras et al. [25] reveals that there is a trade-off between the standard accuracy and adversarial robustness because of the difference between features learned by the optimal standard and optimal robust classifiers. TRADES [32] theoretically analyzes this trade-off. A boundary error is identified between the standard and adversarial error to guide the design of defense against adversarial attacks. As a solution, a tuning parameter $\lambda$ is introduced into their framework to adjust the trade-off. The friendly adversarial training (FAT) [33] generates weak adversarial examples that satisfy a minimal margin of loss. The miss-classified adversarial examples with the lowest classification loss are selected for adversarial training.

### 2.2 Neural Architecture Search

NAS aims to automatically design neural architectures for networks. Early NAS methods [1, 35, 16, 19] are computationally intensive, requiring hundreds or thousands of GPU hours because of the demand of training and evaluation of a large number of architectures. Recently, the differentiable and one-shot NAS approaches Liu et al. [17] and Xu et al. [29] propose to construct a one-shot supernetwork and optimize the architecture parameter with gradient descent, which reduces the computational overhead dramatically. Differentiable neural architecture search allows joint and differentiable optimization of model weights and the architecture parameter using gradient descent. Due to the parallel training of multiple architectures, DARTS is memory consuming. Several follow-up works aim to reduce the memory cost and improve the efficiency of NAS. One remarkable approach among them is PC-DARTS [29]. It utilizes a partial channel connections technique, where sub-channels of the intermediate features are sampled to be processed. Therefore, memory usage and computational cost are reduced. Besides direct reducing the training cost, CARS [30] proposes a novel efficient continuous evolutionary approach based on the historical evaluation. Similarly, PVLL-NAS [14] performs evaluation with a performance estimator, who samples neural architectures for both architecture searching and iterative training of the estimator itself.

Considering adversarial attacks in the optimization of neural architectures can help designing networks that are inherently resistant to adversarial attacks. RACL [5] applied a constraint on the architecture parameter in differentiable one-shot NAS to reduce the Lipschitz constant. Previous works [3, 26] have shown that a smaller Lipschitz constant always corresponds to a more robust network. It is, therefore, effective to improve the robustness of neural architectures by constraining their Lipschitz constant. RobNet [8] directly optimizes the architecture by adversarial training with PGD.

# 3 Methodology

The adversarial training can be considered as a minimax problem, where the adversarial perturbations are generated to attack the network by maximizing the classification loss, and the network is optimized to defend against such attacks:

$$\min_{f} \mathbb{E}_{(\boldsymbol{x},y)\sim\mathcal{D}} \left[ \max_{\boldsymbol{x}'\in B_p(\boldsymbol{x},\varepsilon)} \ell(y, f(\boldsymbol{x}')) \right], \tag{1}$$

where $\mathcal{D}$ is the distribution of the natural examples $\boldsymbol{x}$ and the labels $y$, $B_p(\boldsymbol{x},\varepsilon) = \{\boldsymbol{x}' : \|\boldsymbol{x} - \boldsymbol{x}'\|_p \leq \varepsilon\}$ defines the set of allowed adversarial examples $\boldsymbol{x}'$ within the scale $\varepsilon$ of small perturbations under $l_p$ normalization, and $f$ is the network under attack.

## 3.1 Robust Architecture Dilation

The capacity of deep neural network has been demonstrated to be critical to its adversarial robustness [18, 25, 34]. Madry et al. [18] finds capacity plays an important role in adversarial robustness, and networks require larger capacity for adversarial than standard tasks. Tsipras et al. [25] suggests simple classifier for standard tasks cannot reach good performance on adversarial tasks. However, it remains an open question to use the minimal increase of network capacity in exchange for the adversarial robustness.

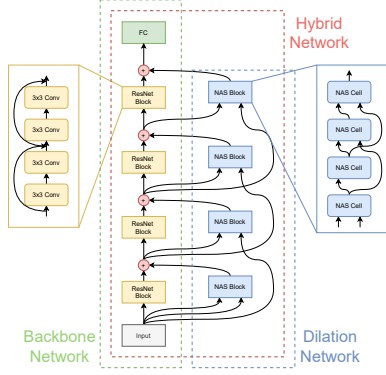

Figure 1: The overall structure of a NADAR hybrid network.

Suppose that we have a backbone network $f_b$ that can achieve a satisfactory accuracy on the natural data. To strengthen its adversarial robustness without hurting the standard accuracy, we propose to increase the capacity of this backbone network $f_b$ by dilating it with a network $f_d$, whose architecture and parameter will be optimized within the adversarial training.

The backbone network $f_b$ is split into blocks. A block $f_b^{(l)}$ is defined as a set of successive layers in the backbone with the same resolution. For a backbone with $L$ blocks, i.e. $f_b = \{f_b^{(l)}, l \in 1, \ldots, L\}$, we attach a cell $f_d^{(l)}$ of the dilation network to each block $f_b^{(l)}$. Therefore, the dilation network also has $L$ cells, i.e. $f_d = \{f_d^{(l)}, l \in 1, \ldots, L\}$. For the dilation architecture, we search for cells within a NASNet-like [35] search space. In a NASNet-like search space, each cell takes two previous outputs as its inputs. The backbone and the dilation network are further aggregated by element-wise sum. The overall structure of a NADAR hybrid network is as shown in Figure 1. Formally, the hybrid network for the adversarial training is defined as:

$$f_{\text{hyb}}(\boldsymbol{x}) = h\left(\odot_{l=1,\ldots,L}\left(f_b^{(l)}(\boldsymbol{z}_{\text{hyb}}^{(l-1)}) + f_d^{(l)}(\boldsymbol{z}_{\text{hyb}}^{(l-1)}, \boldsymbol{z}_{\text{hyb}}^{(l-2)})\right)\right), \tag{2}$$

where $\boldsymbol{z}_{\text{hyb}}^{(l)} = f_b^{(l)}(\boldsymbol{z}_{\text{hyb}}^{(l-1)}) + f_d^{(l)}(\boldsymbol{z}_{\text{hyb}}^{(l-1)}, \boldsymbol{z}_{\text{hyb}}^{(l-2)})$ is the latent feature extracted by the backbone block and the dilation block, and $\odot$ represents functional composition. We also define a classification hypothesis $h : \boldsymbol{z}_{\text{hyb}}^{(L)} \rightarrow \hat{y}$, where $\boldsymbol{z}_{\text{hyb}}^{(L)}$ is the latent representation extracted by the last convolutional layer $L$, and $\hat{y}$ is the predicted label.

During search, the backbone network $f_b$ has a fixed architecture and is parameterized by network weights $\boldsymbol{\theta}_b$. The dilation network $f_d$ is parameterized by not only network weights $\boldsymbol{\theta}_d$ but also the architecture parameter $\boldsymbol{\alpha}_d$. The objective of robust architecture dilation is to optimize $\boldsymbol{\alpha}_d$ for the minimal adversarial loss

$$\min_{\boldsymbol{\alpha}_d} \quad \mathcal{L}_{\text{valid}}^{(\text{adv})}(f_{\text{hyb}}; \boldsymbol{\theta}_d^*(\boldsymbol{\alpha}_d)), \tag{3}$$

$$\text{s.t.} \quad \boldsymbol{\theta}_d^*(\boldsymbol{\alpha}_d) = \operatorname*{argmin}_{\boldsymbol{\theta}_d} \mathcal{L}_{\text{train}}^{(\text{adv})}(f_{\text{hyb}}), \tag{4}$$

where $\mathcal{L}_{\text{train}}^{(\text{adv})}(f_{\text{hyb}})$ and $\mathcal{L}_{\text{valid}}^{(\text{adv})}(f_{\text{hyb}}; \boldsymbol{\theta}_d^*(\boldsymbol{\alpha}_d))$ are the adversarial losses of $f_{\text{hyb}}$ (with the form of Eq. 1) on the training set $\mathcal{D}_{\text{train}}$ and the validation set $\mathcal{D}_{\text{valid}}$, respectively, and $\boldsymbol{\theta}_d^*(\boldsymbol{\alpha}_d)$ is the optimal network weights of $f_d$ depending on the current dilation architecture $\boldsymbol{\alpha}_d$.

## 3.2 Standard Performance Constraint

Existing works on adversarial robustness often fix the network capacity, and the increase of adversarial robustness is accompanied by the standard accuracy drop [25, 32]. However, in this work, we increase

the capacity with dilation, which allows us to increase the robustness while maintaining a competitive standard accuracy. We reach that with a standard performance constraint on the dilation architecture. The constraint is achieved by comparing the standard performance of the hybrid network $f_{\texttt{hyb}}$ to the standard performance of the backbone. We denote the network using the backbone only as $f_{\texttt{bck}}$, which can be formally defined as:

$$f_{\texttt{bck}}(\boldsymbol{x}) = h\left(\odot_{l=1,\dots,L} f_b^{(l)}\left(\boldsymbol{z}_{\texttt{bck}}^{(l-1)}\right)\right), \tag{5}$$

where $\boldsymbol{z}_{\texttt{bck}}^{(l)} = f_b^{(l)}(\boldsymbol{z}_{\texttt{bck}}^{(l-1)})$ is the latent feature extracted by the backbone block. The standard model is optimized with natural examples by:

$$\min_{\boldsymbol{\theta}_b} \quad \mathcal{L}^{(\texttt{std})}(f_{\texttt{bck}}) = \mathbb{E}_{(\boldsymbol{x},y)\sim\mathcal{D}}\left[\ell(f_{\texttt{bck}}(y,\boldsymbol{x}))\right]. \tag{6}$$

where $\mathcal{L}^{(\texttt{std})}$ is the standard loss. Similarly, we can define the standard loss $\mathcal{L}^{(\texttt{std})}(f_{\texttt{hyb}})$ for the hybrid network $f_{\texttt{hyb}}$. In this way, we can compare the two networks by the difference of their losses and constrain the standard loss of the hybrid network to be equal to or lower than the standard loss of the standard network:

$$\mathcal{L}^{(\texttt{std})}(f_{\texttt{hyb}}) - \mathcal{L}^{(\texttt{std})}(f_{\texttt{bck}}) \leq 0. \tag{7}$$

We do not directly optimize the dilation architecture on the standard task, because it is introduced to capture the difference between the standard and adversarial tasks to improve the robustness of the standard trained backbone. It is unnecessary to let both the backbone network and the dilation network to learn the standard task.

### 3.3 FLOPs-Aware Architecture Optimization

By enlarging the capacity of networks, we can improve the robustness, but a drawback is that the model size and computation cost raises. We want to obtain the largest robustness improvement with the lowest computation overhead. Therefore, a computation budget constraint on architecture search is applied. As we are not targeting at any specific platform, the number of **fl**oating point **op**eration**s** (FLOPs) in the architecture instead of the inference latency is considered. The FLOPs is calculated by counting the number of multi-add operations in the network.

We use a differentiable manner to optimize the dilation architecture. In differentiable NAS, a directed acyclic graph (DAG) is constructed as the supernetwork, whose nodes are latent representations and edges are operations. Given that the adversarial training is computationally intensive, to reduce the search cost, a partial channel connections technique proposed by Xu et al. [29] is utilized.

During search, operation candidates for each edge are weighted summed with a softmax distribution of the architecture parameter $\boldsymbol{\alpha}$:

$$\bar{o}^{(i,j)}(\boldsymbol{x}_i) = (1 - S_{i,j}) * \boldsymbol{x}_i + \sum_{o\in\mathcal{O}}\left(\frac{\exp(\boldsymbol{\alpha}_{i,j}^{(o)})}{\sum_{o'\in\mathcal{O}}\exp(\boldsymbol{\alpha}_{i,j}^{(o')})} \cdot o(S_{i,j} * \boldsymbol{x}_i)\right), \tag{8}$$

where $\mathcal{O}$ is a set of operation candidates, $\boldsymbol{x}_i$ is the output of the $i$-th node, and $S_{i,j}$ is binary mask on edge $(i,j)$ for partial channel connections. The binary mask $S_{i,j}$ is set to 1 or 0 to let the channel be selected or bypassed, respectively. Besides the architecture parameter $\boldsymbol{\alpha}$, the partial channel connections technique also introduces a edge normalization weight $\boldsymbol{\beta}$:

$$\boldsymbol{I}^{(j)} = \sum_{i<j}\left(\frac{\exp(\boldsymbol{\beta}_{i,j})}{\sum_{i'<j}\exp(\boldsymbol{\beta}_{i',j})} \cdot \bar{o}^{(i,j)}(\boldsymbol{x}_i)\right), \tag{9}$$

where $\boldsymbol{I}^{(j)}$ is the $j$-th node. The edge normalization can stabilize differentiable NAS by reducing fluctuation in edge selection after search.

Considering Eqs. 8 and 9, the expected FLOPs of the finally obtained discrete architectures from the one-shot supernetwork can be estimated according to $\boldsymbol{\alpha}$ and $\boldsymbol{\beta}$. We calculate the weighted sum of FLOPs of the operation candidates with the identical softmax distributions in Eqs. 8 and 9, which can naturally lead to an expectation. Therefore, the expected FLOPs of node $\boldsymbol{I}^{(j)}$ can be calculated by:

$$\text{FLOPs}(\boldsymbol{I}^{(j)}) = \sum_{i<j}\frac{\exp(\boldsymbol{\beta}_{i,j})}{\sum_{i'<j}\exp(\boldsymbol{\beta}_{i',j})} \cdot \sum_{o\in\mathcal{O}}\frac{\exp(\boldsymbol{\alpha}_{i,j}^{(o)})}{\sum_{o'\in\mathcal{O}}\exp(\boldsymbol{\alpha}_{i,j}^{(o')})} \cdot \text{FLOPs}(o). \tag{10}$$

After that, the FLOPs of the dilation network $\text{FLOPs}(f_d)$ can be estimated by taking the sum of the FLOPs of all the nodes and cells. The objective function in Eq. 3 can be rewritten with the FLOPs constraint as:

$$\min_{\boldsymbol{\alpha}_d} \quad \gamma\log(\text{FLOPs}(f_d))^\tau \cdot \mathcal{L}_{\texttt{valid}}^{(\texttt{adv})}(f_{\texttt{hyb}}), \tag{11}$$

where $\gamma$ and $\tau$ are two coefficient terms. $\tau$ controls the sensitivity of the objective function to the FLOPs constraint, and $\gamma$ scales the constraint to a reasonable range (e.g. around 1.0).

### 3.4 Optimization

We reformulate the bi-level form optimization problem defined in Eqs 3 and 4 into a constrained optimization form. Combining with the standard performance constraint in Eq. 7 and the FLOPs-aware objectives in Eq. 11, we have

$$\min_{\boldsymbol{\alpha}_d} \quad \gamma \log(\text{FLOPs}(f_d))^\tau \cdot \mathcal{L}_{\texttt{valid}}^{(\texttt{adv})}(f_{\texttt{hyb}}; \boldsymbol{\theta}_d^*(\boldsymbol{\alpha}_d)), \tag{12}$$

$$\textbf{s.t.} \quad \mathcal{L}_{\texttt{valid}}^{(\texttt{std})}(f_{\texttt{hyb}}) - \mathcal{L}_{\texttt{valid}}^{(\texttt{std})}(f_{\texttt{bck}}) \leq 0, \tag{13}$$

$$\boldsymbol{\theta}_d^*(\boldsymbol{\alpha}_d) = \operatorname*{argmin}_{\boldsymbol{\theta}_d} \mathcal{L}_{\texttt{train}}^{(\texttt{adv})}(f_{\texttt{hyb}}), \quad \textbf{s.t.} \quad \mathcal{L}_{\texttt{train}}^{(\texttt{std})}(f_{\texttt{hyb}}) - \mathcal{L}_{\texttt{train}}^{(\texttt{std})}(f_{\texttt{bck}}) \leq 0. \tag{14}$$

To solve the constrained architecture optimization problem, we apply a common method for constrained optimization, namely alternating direction method of multipliers (ADMM). To apply ADMM, the objective function needs to be reformulate as an augmented Lagrangian function. We first deal with the upper-level optimization of the architecture parameter $\boldsymbol{\alpha}_d$:

$$L(\{\boldsymbol{\alpha}_d\}, \{\lambda_1\}) = \gamma \log(\text{FLOPs}(f_d))^\tau \cdot \mathcal{L}_{\texttt{valid}}^{(\texttt{adv})}(f_{\texttt{hyb}}) + \lambda_1 \cdot c_1 + \frac{\rho}{2} \| \max\{0, c_1\} \|_2^2 \tag{15}$$

$$\textbf{s.t.} \quad c1 = \mathcal{L}_{\texttt{valid}}^{(\texttt{std})}(f_{\texttt{hyb}}) - \mathcal{L}_{\texttt{valid}}^{(\texttt{std})}(f_{\texttt{bck}}), \tag{16}$$

where $\lambda_1$ is the Lagrangian multiplier, and $\rho \in \mathbb{R}_+$ is a positive number predefined in ADMM. We update $\boldsymbol{\alpha}_d$ and $\lambda_1$ alternately with:

$$\boldsymbol{\alpha}_d^{(t+1)} \leftarrow \boldsymbol{\alpha}_d^{(t)} - \eta_1 \nabla L(\{\boldsymbol{\alpha}_d^{(t)}\}, \{\lambda_1^{(t)}\}) \tag{17}$$

$$\lambda_1^{(t+1)} \leftarrow \lambda_1^{(t)} + \rho \cdot c_1, \tag{18}$$

where $\eta_1$ is a learning rate for architecture. Similarly, the lower-level optimization problem of network weights $\boldsymbol{\theta}_d$ as an augmented Lagrangian function can be defined as:

$$L(\{\boldsymbol{\theta}_d\}, \{\lambda_2\}) = \mathcal{L}_{\texttt{train}}^{(\texttt{adv})}(f_{\texttt{hyb}}) + \lambda_2 \cdot c_2 + \frac{\rho}{2} \| \max\{0, c_2\} \|_2^2 \tag{19}$$

$$\textbf{s.t.} \quad c2 = \mathcal{L}_{\texttt{train}}^{(\texttt{std})}(f_{\texttt{hyb}}) - \mathcal{L}_{\texttt{train}}^{(\texttt{std})}(f_{\texttt{bck}}), \tag{20}$$

where $\lambda_2$ is the Lagrangian multiplier. Similarly, we can update $\boldsymbol{\theta}_d$ and $\lambda_2$ with the same alternate manner:

$$\boldsymbol{\theta}_d^{(t+1)} \leftarrow \boldsymbol{\theta}_d^{(t)} - \eta_2 \nabla L(\{\boldsymbol{\theta}_d^{(t)}\}, \{\lambda_2^{(t)}\}), \tag{21}$$

$$\lambda_2^{(t+1)} \leftarrow \lambda_2^{(t)} + \rho \cdot c_2, \tag{22}$$

where $\eta_2$ is the learning rate for network weights.

## 4 Theoretical Analysis

In this section, we provide theoretical analysis of our proposed NADAR. As there are two major goals in our optimization problem, i.e., the standard performance constraint and the adversarial robustness, this analysis is also twofold. Firstly, a standard error bound of NADAR is analyzed. We demonstrate that the standard error of the dilated adversarial network can be bounded by the standard error of the backbone network and our standard performance constraint. Secondly, we compare the adversarial error of the dilated adversarial network and the standard error of the backbone standard network. We demonstrate that the adversarial performance can be improved by adding a dilation architecture to the backbone, even if the backbone is fixed. These two error bounds can naturally motivate the optimization problem in Eqs. 12 and 13. Detailed proofs are provided in our supplementary material. Besides, through this analysis, we want to reveal two **remarks**: (1) enlarging the backbone network with dilation can improve its performance, which proves the validity of our neural architecture dilation; (2) the dilation architecture should be consistent with the backbone on clean samples and samples that are insensitive to attacks, which directly inspires our standard performance constraint.

We discuss the binary classification case for a simplification, where the label space is $\mathcal{Y} = \{-1, +1\}$. The obtained theoretical results can also be generalized to the multi-class classification case. A binary classification *hypothesis* $h \in \mathcal{H}$ is defined as a mapping $h : \mathcal{X} \mapsto \mathbb{R}$, where $\mathcal{H}$ is a hypothesis space, and $\mathcal{X}$ is an input space of natural examples. The output of the hypothesis is a real value score. The predicted label can be obtained from the score by applying the sign function $\text{sign}(\cdot)$ on it. Denote the backbone hypothesis as $h_b$. By further investigating the influence of the dilation architecture, the hypothesis of the resulting hybrid network can be defined as $h_{\texttt{hyb}}(\boldsymbol{x}) = h_b(\boldsymbol{x}) + h_d(\boldsymbol{x})$, where $h_d$ stands for the change resulting from the dilation architecture. The standard model corresponds to a

hypothesis $h_{\mathtt{bck}}(\boldsymbol{x}) = h_b(\boldsymbol{x})$. We further define the *standard error* of a hypothesis $h$ as
$$R_{\mathtt{std}}(h) := \mathbb{E}\left[\mathbf{1}\{\mathrm{sign}(h(\boldsymbol{x})) \neq y\}\right], \tag{23}$$
and the *adversarial error* of it as
$$R_{\mathtt{adv}}(h) := \mathbb{E}\left[\mathbf{1}\{\exists \boldsymbol{x}' \in B_p(x,\varepsilon), \ \textbf{s.t.} \ \mathrm{sign}(h(\boldsymbol{x}')) \neq y\}\right], \tag{24}$$
where $\mathbf{1}\{\cdot\}$ denotes the indicator function.

## 4.1 Standard Error Bound

To compare the error of two different hypotheses, we first slightly modify the error function. Eq. 23 checks the condition that $\mathrm{sign}(h(\boldsymbol{x})) \neq y$. Because the label space is binary and the output space of $h$ is real value, we can remove the sign function by replacing the condition with $yh(\boldsymbol{x}) \leq 0$. Then, by applying a simple inequality $\mathbf{1}\{yh(\boldsymbol{x}) \leq 0\} \leq e^{-yh(\boldsymbol{x})}$, we have a very useful inequality about the standard error:
$$R_{\mathtt{std}}(h) \leq \mathbb{E}\left[e^{-yh(\boldsymbol{x})}\right]. \tag{25}$$

Eq. 25 can lead to our standard error bound in Theorem 1.

**Theorem 1.** *Let $h_{\mathtt{bck}}(\boldsymbol{x}) = h_b(\boldsymbol{x})$ be a standard hypothesis, $h_{\mathtt{hyb}}(\boldsymbol{x}) = h_b(\boldsymbol{x}) + h_d(\boldsymbol{x})$ be a hybrid hypothesis, and $\mathcal{R}_{\mathtt{std}}(h_{\mathtt{bck}})$ and $\mathcal{R}_{\mathtt{std}}(h_{\mathtt{hyb}})$ be the standard error of $h_{\mathtt{bck}}$ and $h_{\mathtt{hyb}}$, respectively. For any mapping $h_b, h_d : \mathcal{X} \mapsto \mathbb{R}$, we have*
$$\mathcal{R}_{\mathtt{std}}(h_{\mathtt{hyb}}) \leq \mathcal{R}_{\mathtt{std}}(h_{\mathtt{bck}}) + \mathbb{E}\left[e^{-h_b(\boldsymbol{x})h_d(\boldsymbol{x})}\right], \tag{26}$$
*where $\boldsymbol{x} \in \mathcal{X}$ is the input.*

Theorem 1 illustrates that the standard performance of the hybrid network is bounded by the standard performance of the backbone network and the sign disagreement between $h_b(\boldsymbol{x})$ and $h_d(\boldsymbol{x})$. This reflects our **remark (2)**. If the backbone accurately predicts the label of the natural data $x$, $h_d(\boldsymbol{x})$ shall make the same category prediction, which implies that the prediction by the hybrid hypothesis $h_{\mathtt{hyb}}(\boldsymbol{x}) = h_b(\boldsymbol{x}) + h_d(\boldsymbol{x})$ can be strengthened and would not lead to a worse result than that of the stand hypothesis. To reach such objective, it naturally links with the standard performance constraint proposed in Eq. 7 and applied in Eq. 13.

## 4.2 Adversarial Error Bound

Similar to Eq. 25, we can have an inequality about the adversarial error:
$$R_{\mathtt{adv}}(h) \leq \mathbb{E}\left[\max_{\boldsymbol{x}' \in B_p(x,\varepsilon)} e^{-yh(\boldsymbol{x}')}\right], \tag{27}$$
based on which we can derive the following Lemma 2.

**Lemma 2.** *For any mapping $h : \mathcal{X} \mapsto \mathbb{R}$, we have*
$$\mathbb{E}\left[\max_{\boldsymbol{x}' \in B_p(x,\varepsilon)} e^{-yh(\boldsymbol{x}')}\right] \leq \mathbb{E}\left[\max_{\boldsymbol{x}' \in B_p(x,\varepsilon)} e^{-yh(\boldsymbol{x})}e^{-h(\boldsymbol{x})h(\boldsymbol{x}')}\right], \tag{28}$$
*where $\boldsymbol{x} \in \mathcal{X}$ is the input, $y \in \{-1,+1\}$ is the corresponding label, and $\varepsilon$ is the bound of allowed adversarial perturbation.*

Lemma 2 is an inherent feature of a single hypothesis. We generalize it to the case of dilating $h_{\mathtt{bck}}$ to $h_{\mathtt{hyb}}$ with a dilation hypothesis $h_d$.

**Theorem 3.** *Let $h_{\mathtt{bck}}(\boldsymbol{x}) = h_b(\boldsymbol{x})$ be a standard hypothesis, $h_{\mathtt{hyb}}(\boldsymbol{x}) = h_b(\boldsymbol{x}) + h_d(\boldsymbol{x})$ be a dilated hypothesis, $\mathcal{R}_{\mathtt{std}}(h_{\mathtt{bck}})$ be the standard error of $h_{\mathtt{bck}}$, and $\mathcal{R}_{\mathtt{adv}}(h_{\mathtt{hyb}})$ be the adversarial error of $h_{\mathtt{hyb}}$. For any mapping $h_b, h_d : \mathcal{X} \mapsto \mathbb{R}$, we have*
$$\mathcal{R}_{\mathtt{adv}}(h_{\mathtt{hyb}}) \leq \mathcal{R}_{\mathtt{std}}(h_{\mathtt{bck}}) + \mathbb{E}\left[\max_{\boldsymbol{x}' B_p(x,\varepsilon)} e^{-yh_b(\boldsymbol{x})}\left(e^{-h_b(\boldsymbol{x})h_b(\boldsymbol{x}')}e^{-yh_d(\boldsymbol{x}')} - 1\right)\right]. \tag{29}$$
*where $\boldsymbol{x} \in \mathcal{X}$ is the input, $y \in \{-1,+1\}$ is the corresponding label, and $\varepsilon$ is the bound of allowed adversarial perturbation.*

By minimizing $e^{-yh_b(\boldsymbol{x})}$ in Theorem 3, we expect the backbone network to have a satisfactory accuracy on the natural data, which is a prerequisite of the proposed algorithm. As the backbone network has been fixed in this paper, the term $e^{-h_b(\boldsymbol{x})h_b(\boldsymbol{x}')}$ will not be influenced by the algorithm. The remaining term $e^{-yh_d(\boldsymbol{x}')}$ implies that even if the backbone network makes wrong prediction on the adversarial example $\boldsymbol{x}'$, there is still a chance for the dilation network $h_d$ to correct the mis-classification and improve the overall adversarial accuracy of the hybrid network $h_{\mathtt{hyb}}$. This capability of dilation reflects our **remark (1)**. In another case, if $h_b$ makes a correct prediction, $h_d$ should agree with it, which reflects our **remark (2)** is also applied to the adversarial error.

Table 1: The standard validation accuracy on natural images and adversarial validation accuracy under various attacks of NADAR comparing to different SOTA methods on CIFAR-10.

| Category | Method | Params (M) | | +× (G) | | Valid Acc. Against (%) | | | | |
|---|---|---|---|---|---|---|---|---|---|---|
| | | Back. | Arch. | Back. | Arch. | Natural | FGSM | PGD-20 | PGD-100 | MI-FGSM |
| Standard | Standard | 46.2 | - | 6.7 | - | 95.01 | 0.00 | 0.00 | 0.00 | 0.00 |
| Adversarial Training | PGD-7 [18] | 46.2 | - | 6.7 | - | 87.25 | 56.10 | 45.84 | 45.29 | - |
| | FAT [33] | 46.2 | - | 6.7 | - | 89.34 | 65.52 | 46.13 | 46.82 | - |
| | FreeAT-8 [22] | 46.2 | - | 6.7 | - | 85.96 | - | 46.82 | 46.19 | - |
| | TRADES-1 [32] | 46.2 | - | 6.7 | - | 88.64 | - | 48.90 | - | 51.26 |
| | TRADES-6 [32] | 46.2 | - | 6.7 | - | 84.92 | - | 56.43 | - | 57.95 |
| Standard NAS | AmoebaNet [20] | - | 3.2 | - | 0.5 | 83.41 | 56.40 | 39.47 | - | 47.60 |
| | NASNet [35] | - | 3.8 | - | 0.6 | 83.66 | 55.67 | 48.02 | - | 53.05 |
| | DARTS [17] | - | 3.3 | - | 0.5 | 83.75 | 55.75 | 44.91 | - | 51.63 |
| | PC-DARTS [29] | - | 3.6 | - | 0.6 | 83.94 | 52.67 | 41.92 | - | 49.09 |
| Robust NAS | RobNet-small [8] | - | 4.4 | - | N/A | 78.05 | 53.93 | 48.32 | 48.08 | 48.98 |
| | RobNet-medium [8] | - | 5.7 | - | N/A | 78.33 | 54.55 | 49.13 | 48.96 | 49.34 |
| | RobNet-large [8] | - | 6.9 | - | N/A | 78.57 | 54.98 | 49.44 | 49.24 | 49.92 |
| | RACL [5] | - | 3.6 | - | 0.5 | 83.89 | 57.44 | 49.34 | - | 54.73 |
| Dilation | NADAR-A (ours) | 46.2 | 3.6 | 6.7 | 0.6 | 86.61 | 59.98 | 52.84 | 52.54 | 57.72 |
| | NADAR-B (ours) | 46.2 | 4.4 | 6.7 | 0.7 | 86.23 | 60.46 | 53.43 | 53.06 | 58.43 |

# 5 Experiments

We perform extensive experiments to demonstrate that NADAR can improve the adversarial robustness of neural networks by dilating the neural architecture. In this section, we first compare both the standard and adversarial accuracy of our hybrid network to various state-of-the-art (SOTA) methods. Then, we perform experiments to analyze the impact of each component in the NADAR framework, including the dilation-based training approach and the standard performance constraint. Finally, we explore the sufficient scale of dilation and the effect of FLOPs constraint. More results on other datasets under various attacking manners with different backbones are also available in the supplementary material.

## 5.1 Experiment Setting

We use a similar pipeline to previous NAS works [17, 29, 5, 8]. Firstly, we optimize the dilating architecture in a one-shot model. Then, a discrete architecture is derived according to the architecture parameters $\alpha$ and $\beta$. Finally, a discrete network is constructed and retrained for validation. During the dilation phase, the training set is split into two equal parts. One is used as the training set for network weights optimization, and the other one is used as the validation set for architecture parameter optimization. During the retraining and validation phases, the entire training set is used for training, and the trained network is validated on the original validation set.

We perform dilation under white-box attacks on CIAFR-10/100 [10] and ImageNet [21] and under black-box attacks on CIFATR-10. The NADAR framework requires a backbone to be dilated. Following previous works [18, 22, 33, 32], we use the 10 times wider variant of ResNet, i.e. the Wide ResNet 34-10 (WRN34-10) [31], on both CIFAR dataset, and use ResNet-50 [9] on ImageNet. The search space of dilated architecture and the dilated architectures are as illustrated in Section A of the supplementary material.

Considering both the optimization of neural architecture and the generation of adversarial examples is computational intensive, we apply methods to reduce the computational overhead during the dilation phase. As aforementioned, we utilize partial channel connections to reduce the cost of architecture optimization. As for the adversarial training during search, we use FreeAT [22], which recycles gradients during training for the generation of adversarial examples and reduces the training cost.

## 5.2 Defense Against White-box Attacks

**CIFAR-10.** We compare the hybrid network with 4 categories of SOTA methods, including standard training, adversarial training, standard NAS, and robust NAS. The standard training method and all the adversarial training methods use the WRN34-10. For the standard NAS methods, the architecture is the best architecture searched with standard training as reported in their papers and is retrained with PGD-7 [18]. For the adversarial NAS methods, we follow their original setting. We include two best dilation architectures obtained with our method, NADAR-A and NADAR-B dilated without and with the FLOPs constraint, respectively. The architectures are visualized in Section A of the supplementary material. Our architectures are also retrained with PGD-7. In Table 1, the standard accuracy on natural images and the adversarial accuracy under PGD-20 attack are reported.

Comparing NADAR-A and NADAR-B, the FLOPs constraint can obviously reduce the FLOPs number (reducing by 14.28%) as well as the parameters number (reducing by 18.19%) of the dilation

Table 2: The standard and adversarial validation on CIFAR-100.

| Method | Valid Acc. Against (%) | |
|---|---|---|
| | Natural | PGD-20 |
| Standard | 78.84 | 0.00 |
| PGD-7 [18] | - | 23.20 |
| FreeAT-8 [22] | 62.13 | 25.88 |
| RobNet-large [8] | - | 23.19 |
| RACL [5] | - | 27.80 |
| NADAR-A (ours) | 61.73 | 27.77 |
| NADAR-B (ours) | 62.56 | 28.40 |

Table 3: The standard and adversarial validation accuracy on Tiny-ImageNet with ResNet-50 as backbone.

| Architecture | Training Method | GPU Days | Valid Acc. Against (%) | | | |
|---|---|---|---|---|---|---|
| | | | Natural | FGSM | PGD-10 | PGD-20 |
| Backbone | PGD-4 | 0.19 | 43.23 | 24.13 | 22.25 | 22.16 |
| Dilation (ours) | PGD-4 | 0.45 | 44.37 | 24.33 | 22.69 | 22.70 |
| Backbone | FreeAT-4 | 0.05 | 42.73 | 24.10 | 22.67 | 22.58 |
| Dilation (ours) | FreeAT-4 | 0.12 | 44.68 | 24.66 | 22.88 | 22.78 |
| Backbone | FastAT | 0.12 | 45.92 | 23.53 | 20.66 | 20.54 |
| Dilation (ours) | FastAT | 0.22 | 46.22 | 23.90 | 21.21 | 21.14 |

architecture. The negative impact of it on the adversarial accuracy is marginal (only 0.59% under PGD-20 attack), and the standard accuracy can even be slightly improved.

As for adversarial training methods, we improve the adversarial performance by 7.59% with only 1.02% standard performance drop comparing to PGD-7 while we use the same training method but the hybrid network. This result illustrates that our dilation architecture can indeed improve the robustness of a network without modifying the training method. At the meantime, the standard accuracy is constrained to a competitive level. Comparing to FreeAT-8, which is their best adversarial setting, our method reaches both lower standard accuracy drop and higher adversarial accuracy gain than it.

A very different method to PGD and FreeAT, namely friendly adversarial training (FAT), aims to improve the standard accuracy of adversarially trained models by generating weaker adversarial examples than regular adversarial training. Although FAT can make significant improvement on standard accuracy with weak attack, its adversarial accuracy gain against PGD-7 is marginal (only 0.29%). It even has lower adversarial accuracy than FreeAT-8 despite its higher standard accuracy. Unlike FAT, our method is dedicated to another direction of improvement, which improves the adversarial robustness without significantly affecting the standard performance. Even though there is still a trade-off between the standard and the adversarial accuracy, we can increase the ratio of the standard drop to the adversarial gain to $1 : 7.44$.

A previous work focuses on the trade-off is TRADES, which introduces a tuning parameter ($\lambda$) to adjust the balance of the trade-off. Nevertheless, comparing the TRADES-1 ($1/\lambda = 1$) and TRADES-6 ($1/\lambda = 6$), their trade-off ratio is only $1 : 2.02$ (i.e. $3.72\%$ standard accuracy drop for $7.53\%$ adversarial accuracy gain). We can provide a better ratio of trade-off than them. Besides, our standard performance is naturally constrained by Eq. 7. There are no hyperparameters in it that needs to be adjusted, which leads to our better trade-off ratio of the standard drop to the adversarial gain and more reasonable balance than TRADES.

Finally, comparing to NAS methods, our hybrid network can outperform both the standard and robust NAS architectures. The standard NAS architectures are not optimized for adversarial robustness. Except the NASNet, their adversarial accuracies are generally poor. Although they are optimized for standard tasks, their standard accuracy after adversarial training is significantly lower than the WRN34-10 trained with PGD-7. As for robust NAS methods, RobNet significantly sacrifices their standard accuracy for robustness, which has the lowest standard accuracy among all the works listed in Table 1. RACL has a better trade-off, but it can only reach the standard accuracy of standard NAS architecture, which is still lower than adversarilly trained WRN34-10. This demonstrates that dilating a standard backbone for both standard constraint and adversarial gain is more effectiveness than design a new architecture from scratch.

**CIFAR-100.** We adapt architectures dilated on CIFAR-10 to CIFAR-100, and report the results in Table 2. We consider two kinds of baselines, including traditional adversarial training methods (PGD-7 and FreeAT-8) and two robust NAS methods (RobNet and RACL). The results show that even with more categories, NADAR can still reach superior robustness under PGD-20 attack. As for the standard validation accuracy, we can reach competitive performance comparing to FreeAT-8. The other works do not report standard accuracy in their papers. As for the adversarial validation accuracy, our NADAR-B can outperform all the baselines, while NADAR-A is slightly lower than RACL but significantly better than the others.

**Tiny-ImageNet.** We also adapt our architectures to a larger dataset, namely Tiny-ImageNet. For efficient training on Tiny-ImageNet, we compare our dilated architecture with PGD and two efficient adversarial training method, i.e. FreeAT [22] and FastAT [28]. We follow the ImageNet setting of Shafahi et al. [22] and Wong et al. [28], which uses ResNet-50 as the backbone and set the clip size $\epsilon = 4$. For PGD and FreeAT, we set the number of steps $K = 4$ and the step size $\epsilon_S = 2$ The results are reported in Table 3. We also report the GPU days cost to train the networks with NVIDIA V100

GPU. Although NADAR consumes approximately 1.8∼2.4× GPU days, our method can consistently outperform the baselines in terms of both natural and adversarial accuracy.

## 5.3 Defense Against AutoAttack

Beside the traditional attack methods, we also consider a novel and promising parameter-free evaluation method, namely AutoAttack [4]. We use the standard setting of AutoAttack, including four individual attacks: $APGD_{CE}$, $APGD_{DLR}^{T}$, $FAB^{T}$ and Square. The column AA is a combination of the four attacks. The validation accuracy are reported in Table 4. As can be seen, our method reaches superior performance

Table 4: The adversarial validation accuracy of NADAR comparing to different SOTA methods under AutoAttack on CIFAR-10.

| Category | Method | Valid Acc. Against (%) | | | | |
|---|---|---|---|---|---|---|
| | | $APGD_{CE}$ | $APGD_{DLR}^{T}$ | $FAB^{T}$ | Square | AA |
| Adversarial Training | PGD-7 [18] | 44.75 | 44.28 | 44.75 | 53.10 | 44.04 |
| | FastAT [28] | 45.90 | 43.22 | 43.74 | 53.32 | 43.21 |
| | FreeAT-8 [22] | 43.66 | 41.64 | 43.44 | 51.95 | 41.47 |
| Dilation | NADAR-A (ours) | 52.27 | 50.00 | 50.00 | 58.69 | 49.83 |
| | NADAR-B (ours) | 52.64 | 50.45 | 50.88 | 59.33 | 50.44 |

than the baselines. For simplification, we only list the comparison to the best performance among PGD-7, FastAT and FreeAT-8 as follow: under $APGD_{CE}$ attack, we can outperform FastAT by 6.74%; under $APGD_{DLR}^{T}$, we can outperform PGD-7 by 6.17%; under $FAB^{T}$ attacks we can outperform PGD-7 by 6.13%; under Square attack, we can outperform FastAT by 6.01%.

## 5.4 Defense Against Black-box Attacks

We perform black-box attacks on CIFAR-10. We use different source networks to generate adversarial examples. For the source networks, we use the WRN34-10 backbone trained with natural images and adversarial images generated with FGSM and PGD-7. For the defense networks, we compare our best NADAR-B architec-

Table 5: The adversarial validation accuracy under black-box attacks on CIFAR-10.

| Defense Network | Source Network | Valid Acc. (%) | | | |
|---|---|---|---|---|---|
| | | FGSM | PGD-20 | PGD-100 | MI-FGSM |
| WRN34-10 + PGD-7 | WRN34-10 + Natural | 83.99 | 84.56 | 84.76 | 84.05 |
| NADAR-B + PGD-7 | WRN34-10 + Natural | 85.94 | 86.59 | 86.51 | 85.95 |
| WRN34-10 + PGD-7 | WRN34-10 + FGSM | 70.78 | 68.26 | 68.30 | 69.73 |
| NADAR-B + PGD-7 | WRN34-10 + FGSM | 77.25 | 77.66 | 77.69 | 77.19 |
| WRN34-10 + PGD-7 | NADAR-B + PGD-7 | 69.33 | 67.08 | 67.11 | 68.26 |
| NADAR-B + PGD-7 | WRN34-10 + PGD-7 | 70.78 | 68.26 | 68.30 | 69.73 |

ture with the plain WRN34-10 backbone. Both of them are trained with PGD-7. The results are reported in Table 5 grouped according to source networks. With the WRN34-10 source network trained with natural images and FGSM, NADAR-B can consistently outperform the backbone. We also use NADAR-B trained with PGD-7 and WRN34-10 trained with PGD-7 to attack each other. Our hybrid network can consistently reach superior performances.

## 5.5 NADAR Trained with Different Adversarial Training Methods

In previous experiments, our NADAR is trained with PGD-7, and other competitors are trained according to their corresponding settings. To further investigate whether NADAR can work along with other stronger adversarial training methods than the PGD-7, we train the obtained architecture with various adversarial training methods, including FAT, TRADES-1, and TRADES-6. The results are shown in Table 6. As can be seen, our method can consistently outperform the backbone in terms of adversarial accuracy. Regarding natural accuracy, NADAR is competitive to the backbone (only 0.52% lower

Table 6: Comparison of test accuracy of NADAR and WRN34-10 backbone when using various AT methods for training.

| Model | AT Method | Valid Acc. Against (%) | | |
|---|---|---|---|---|
| | | Natural | PGD-20 | AA |
| Backbone | PGD-7 | 87.25 | 45.84 | 44.04 |
| NADAR (ours) | PGD-7 | 86.23 | $53.43^{(+7.59)}$ | $50.44^{(+6.4)}$ |
| Backbone | FAT | 89.34 | 46.13 | N/A |
| NADAR (ours) | FAT | 88.12 | $54.53^{(+8.40)}$ | $51.37^{(N/A)}$ |
| Backbone | TRADES-1 | 88.64 | 48.90 | 43.01 |
| NADAR (ours) | TRADES-1 | 89.77 | $55.13^{(+6.23)}$ | $50.90^{(+7.89)}$ |
| Backbone | TRADES-6 | 84.92 | 56.43 | 53.08 |
| NADAR (ours) | TRADES-6 | 83.94 | $57.43^{(+1.00)}$ | $55.25^{(+2.17)}$ |

in average, which is a marginal drop, given the magnitude of robustness improvement). Especially, under the TRADES-1 setting, which focuses on natural accuracy, NADAR can outperform its backbone on both natural accuracy (89.77% vs. 88.64%) and adversarial accuracy (55.13% vs. 48.90% under PGD-20 and 50.90% vs. 43.01% under AutoAttack). In the meantime, its adversarial accuracy (55.13%) is much closer to the one of TRADES-6 (56.43%), which focuses on robustness and has low (4.85% lower) natural accuracy, than to the one of TRADES-1 (48.90%), which has a similar natural accuracy.

## 5.6 Ablation Study of Dilation Method

We perform ablation study of the dilation method. There are two crucial components in our method. Firstly, the separate optimization objectives of standard and adversarial tasks can ensure the that the backbone focus on clear images, and the dilation network learns to improve the robustness of the backbone. Secondly, the standard performance constraint prevents the dilation network from

harming the standard performance of the backbone network. This experiment demonstrates that both of them make crucial contributions to the final results. Note that without the separate objectives, the hybrid network is trained as a whole. Therefore, there is also no standard constraint. We use the same settings to Section 5.2.

The standard and adversarial accuracy of the obtained networks by retraining is reported in Table 7. If there is no standard performance constraint, dilating together or separately has the similar standard performance. Although the backbone of the latter is trained with standard objective, it won't influence the retraining results too much (only 0.6% higher in average). The complete framework with standard constraint reaches the best standard accuracy after

Table 7: The standard and adversarial accuracy by retraining of various networks dilated with ablated manners.

| Separate Objectives | Standard Constraint | Valid Acc. Against (%) | |
|---|---|---|---|
| | | Natural | PGD-20 |
| No | N/A | 84.19±0.32 | 45.97±0.18 |
| Yes | No | 84.79±0.55 | 48.53±0.33 |
| Yes | Yes | 85.97±0.26 | 53.18±0.25 |

retraining. As for the adversarial accuracy, dilating with the separate objectives can consistently outperforms dilating with a single adversarial objectives.

# 6 Conclusion

The trade-off between accuracy and robustness is considered as an inherent property of neural networks, which cannot be easily bypassed with adversarial training or robust NAS. In this paper, we propose to dilate the architecture of neural networks to increase the adversarial robustness while maintaining a competitive standard accuracy with a straightforward constraint. The framework is called neural architecture dilation for adversarial robustness (NADAR). Extensive experiments demonstrate that NADAR can effectively improve the robustness of neural networks and can reach a better trade-off ratio than existing methods.

## Acknowledgments

The authors would like to thank the area chairs and the reviewers for their constructive comments. This work was supported in part by the Australian Research Council under projects DE180101438 and DP210101859.

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
