# Neural Architecture Dilation for Adversarial Robustness (Supplementary Material)

**Yanxi Li** [1], **Zhaohui Yang** [2,3], **Yunhe Wang** [2], **Chang Xu** [1]

[1] School of Computer Science, University of Sydney, Australia
[2] Huawei Noah's Ark Lab
[3] Key Lab of Machine Perception (MOE), Department of Machine Intelligence, Peking University, China
yali0722@uni.sydney.edu.au, zhaohuiyang@pku.edu.cn,
yunhe.wang@huawei.com, c.xu@sydney.edu.au

## A    Search Space and Dilated Architectures

For the dilation architecture, we use a DAG with 4 nodes as the supernetwork. There are 8 operation candidates for each edges, including 4 convolutional operations: $3 \times 3$ separable convolutions, $5 \times 5$ separable convolutions, $3 \times 3$ dilated separable convolutions and $5 \times 5$ dilated separable convolutions, 2 pooling operations: $3 \times 3$ average pooling and $3 \times 3$ max pooling, and two special operations: an identity operation representing skip-connection and a zero operation representing two nodes are not connected. During dilating, we stack 3 cells for each of the 3 blocks in the WRN34-10. During retraining, the number is increased to 6.

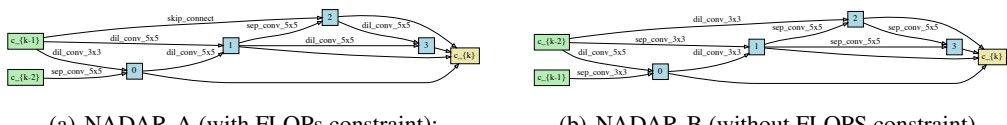

(a)  NADAR-A (with FLOPs constraint);     (b)  NADAR-B (without FLOPS constraint).

Figure 1: Visualization of the dilated cells.

The dilated architectures designed by NADAR are as shown in Figure 1. We find that NADAR prefers deep architecture, which can increase more non-linearity with limited number of parameters. The non-linearity is closely related to network capacity. Such deep architectures can bring more capacity and adversarial robustness to the hybrid network.

## B    Additional Results

### B.1    MNIST

Despite adaptation, we report the adversarial validation accuracy of architectures dilated by NADAR under various attack methods on MNIST and a colorful variant of MNIST, namely MNIST-M [2]. MNIST-M blends greyscale images in MNIST over random patches of colour photos in BSDS500 [1]. The blending introduces extra colour and texture. In this experiment, we use the ResNet-18 as our backbone.

Table 1 shows the adversarial validation accuracy. We report the results of NADAR with and without FLOPs constraint. As shown in the table, the FLOPs constraint can reduce the FLOPs by $20.75 \sim 35.72\%$, while the performance is still competitive. On the MNIST, we observe that NADAR can reach better accuracy under PGD-40 than under FGSM and MI-FGSM. We argue that this is because the MNIST dataset is relatively simple, and the 40 steps PGD causes overfitting. We therefore perform experiments on the MNIST-M, and the results shows that FGSM > MI-FGSM > PGD-40.

35th Conference on Neural Information Processing Systems (NeurIPS 2021).

Table 1: The adversarial validation accuracy of NADAR under the FGSM, MI-FGSM, and PGD-40 attack on MNIST and MNIST-M.

| Dataset | FLOPs Const. | +× (M) | Valid Acc. Against (%) | | |
|---|---|---|---|---|---|
| | | | FGSM | MI-FGSM | PGD-40 |
| MNIST | T | 104.02 | 98.19 | 98.11 | 98.90 |
| | F | 131.26 | 98.27 | 98.25 | 98.97 |
| MNIST-M | T | 89.23 | 92.50 | 92.31 | 91.79 |
| | F | 138.81 | 93.47 | 93.04 | 92.62 |

We also compare the results to SOTA methods. On the MNIST dataset, PGD-7 only reaches 96.01% validation accuracy under PGD-40 attack, and TRADES-6 only reaches 96.07%.

### B.2 Dilation with Various Backbones

To demonstrate the generalizability of NADAR, we test it with different scale of ResNet backbones from ResNet-18 to ResNet-101. The standard accuracy and adversarial accuracy under PGD-20 attack of various backbones are reported in Table 2. All the hybrid networks are retrained with PGD-7 as the same setting in our paper.

Table 2: NADAR with various backbones.

| Network | Valid Acc. Against (%) | |
|---|---|---|
| | Natural | PGD-20 |
| WRN34-10 w/o dilation | 87.25 | 45.84 |
| ResNet-18 + NADAR | 81.35 | 50.92 |
| ResNet-34 + NADAR | 83.57 | 52.64 |
| ResNet-50 + NADAR | 83.23 | 52.89 |
| ResNet-101 + NADAR | 84.39 | **53.89** |
| WRN34-10 + NADAR | **86.23** | 53.43 |

The results demonstrate that NADAR can effectively improve the robustness of different backbones comparing to the WRN34-10 baseline without any dilation. The largest ResNet-101 backbone can reach competitive adversarial accuracy to the dilated WRN34-10. Regarding standard accuracy on natural images, all the ResNet backbones suffer higher performance drop comparing to the WRN34-10 backbone due to the inherent limitation of the small capacity of the backbone itself. This illustrates that although NADAR can improve the robustness regardless of the capacity of the backbone, it is still crucial to select a proper backbone for better standard accuracy.

## C Additional Ablation Studies

### C.1 Adversarial Training for Dilation

As aforementioned, we use the FressAT as the adversarial training method to optimize the dilation architecture for efficiency. FreeAT requires a repeat number $K$ on each mini-batch for better perturbation generation. According to their paper, $K = 8$ reaches the best robustness. We also perform experiments regarding the selection of $K$. Fig. 2 illustrates the accuracy curves of the hybrid network during dilating. We report the adversarial training accuracy of FreeAT, the standard validation accuracy, and the adversarial validation accuracy under PGD-20 attacks. There is no standard training accuracy, because the hybrid network is not directly optimized under the standard classification task (recall the standard performance constraint). All the values are obtained after each complete epochs, when the $K$-repeat of all the mini-batches are finished. The horizontal axis represents the total number of optimization steps, which equals to the epoch number multiply by $K$.

When $K = 4$, the hybrid network reaches outstanding adversarial training accuracy, but the validation only increases slightly at the very beginning of training, and then keep decreasing until reach 0. In the contrast, the standard validation accuracy increases continuously and reaches a competitive level. This implies that the perturbation generated with $K = 4$ is not powerful enough to dilate the network for the defense against PGD-20, and the framework might be dominated by the standard training

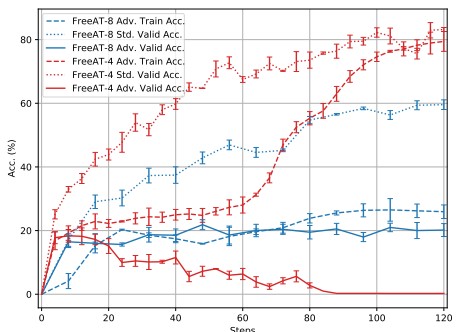

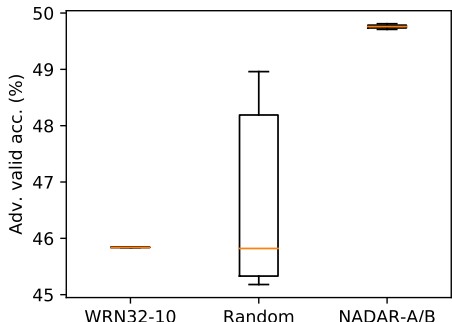

Figure 2: The accuracy curves of dilating architectures with different adversarial training settings of FreeAT.

Figure 3: Comparison of NADAR to WRN34-10 backbone and randomly dilated hybrid networks.

Table 3: Different number of stacked cells in the dilation network.

| # Dilation Cells | Params (M) | | +× (G) | | Valid Acc. Against (%) | |
|---|---|---|---|---|---|---|
| | Back. | Arch. | Back. | Arch. | Natural | PGD-20 |
| $3 \times 3$ | 46.2 | 2.0 | 6.7 | 0.3 | 86.45±0.22 | 47.78±0.41 |
| $3 \times 6$ | 46.2 | 4.4 | 6.7 | 0.7 | 86.28±0.26 | 49.63±0.18 |
| $3 \times 9$ | 46.2 | 6.8 | 6.7 | 1.1 | 85.63±0.12 | 45.25±0.57 |

of the backbone or the standard constraint on the dilation architecture. When $K = 8$, although the standard validation accuracy is much lower than the previous result, the adversarial training and validation accuracy is competitive and close to each other.

### C.2 Different Scales of Dilation

Beside the FLOPs constraint, there is another factor that impacts the model capacity and the computation cost of a dilation network, that is the number of stacked cells. In Section 5.2, we stack 6 cells for each of the 3 blocks in the WRN34-10 for retraining. Intuitively, large network capacity corresponds to better performance. However, we demonstrate that the network cannot be dilate unlimitedly. There is a sweet spot of neural architecture dilation. In this experiment, we test two more scales of stacked cells. Table 3 compares the validation results of different scales of dilation.

We can observe that as the scale of dilation network increases, the standard accuracy consistently declines. In terms of the adversarial accuracy, it first increases with the dilation scale, and then drops significantly. This might because the network becomes difficult to converge as the network capacity increases. Therefore, we stack $3 \times 6$ cells in the dilation network, which reaches the best adversarial accuracy and has a lower standard accuracy drop.

### C.3 Comparison to Random Dilation

To demonstrate the effectiveness of neural architecture dilation, we compare five randomly dilated architectures to our NADAR architectures and the WRN34-10 backbone. We train all the networks with PGD-7 and test their robustness under PGD-20. The adversarial validation accuracy is as shown in Figure 3. The median accuracy of random architectures is similar to the WRN34-10 backbone, but with a great possibility to reach better performance. However, there is still a certain possibility that the dilation architectures can slightly harm the performance of the hybrid network. This shows that neural architecture dilation has the potential to improve the robustness of a backbone, but it still needs to be optimized. The accuracy of NADAR-A and -B is significantly better than the best results of random dilation, which shows that our approach can indeed improve the robustness of backbones effectively and stably.

## D Proof of Theorems

This section proves the lemmas and theorems in our paper.

### D.1 Standard Error Bound

**Theorem 1.** *Let $h_{\text{bck}}(\boldsymbol{x}) = h_b(\boldsymbol{x})$ be a standard hypothesis, $h_{\text{hyb}}(\boldsymbol{x}) = h_b(\boldsymbol{x}) + h_d(\boldsymbol{x})$ be a hybrid hypothesis, and $\mathcal{R}_{\text{std}}(h_{\text{bck}})$ and $\mathcal{R}_{\text{std}}(h_{\text{hyb}})$ be the standard error of $h_{\text{bck}}$ and $h_{\text{hyb}}$, respectively. For any mapping $h_b, h_d : \mathcal{X} \mapsto \mathbb{R}$, we have*

$$\mathcal{R}_{\text{std}}(h_{\text{hyb}}) \leq \mathcal{R}_{\text{std}}(h_{\text{bck}}) + \mathbb{E}\left[e^{-h_b(\boldsymbol{x})h_d(\boldsymbol{x})}\right], \tag{1}$$

*where $\boldsymbol{x} \in \mathcal{X}$ is the input.*

*Proof.* In Theorem 1, we compare the standard error $\mathcal{R}_{\text{std}}$ of $h_{\text{bck}}$ and $h_{\text{hyb}}$. The error bound can be defined as the disagreement between the two hypothesis under the condition that $h_{\text{bck}}$ is correct. Formally, it can be written as

$$\mathcal{R}_{\text{std}}(h_{\text{adv}}) - \mathcal{R}_{\text{std}}(h_{\text{bck}}) \tag{2}$$
$$= \mathbb{E}\left[\mathbf{1}\left(yh_{\text{bck}}(\boldsymbol{x}) > 0,\ h_{\text{hyb}}(\boldsymbol{x})h_{\text{bck}}(\boldsymbol{x}) \leq 0\right)\right] \tag{3}$$
$$\leq \mathbb{E}\left[\mathbf{1}\left(h_{\text{hyb}}(\boldsymbol{x})h_{\text{bck}}(\boldsymbol{x}) \leq 0\right)\right]. \tag{4}$$

By applying a simple inequality

$$\mathbf{1}\{yh(\boldsymbol{x}) \leq 0\} \leq e^{-yh(\boldsymbol{x})}, \tag{5}$$

we have:

$$\mathbb{E}\left[\mathbf{1}\left(h_{\text{hyb}}(\boldsymbol{x})h_{\text{bck}}(\boldsymbol{x}) \leq 0\right)\right] \tag{6}$$
$$\leq \mathbb{E}\left[e^{-h_{\text{hyb}}(\boldsymbol{x})h_{\text{bck}}(\boldsymbol{x})}\right] \tag{7}$$
$$= \mathbb{E}\left[e^{-(h_b(\boldsymbol{x})+h_w(\boldsymbol{x}))h_b(\boldsymbol{x})}\right] \tag{8}$$
$$= \mathbb{E}\left[e^{-h_b(\boldsymbol{x})h_b(\boldsymbol{x})}e^{-h_b(\boldsymbol{x})h_w(\boldsymbol{x})}\right]. \tag{9}$$

As $h_b(\boldsymbol{x})h_b(\boldsymbol{x}) \in [0, +\infty)$, we have $e^{-h_b(\boldsymbol{x})h_b(\boldsymbol{x})} \in (0, 1]$. Therefore, we have

$$\mathcal{R}_{\text{std}}(h_{\text{hyb}}) - \mathcal{R}_{\text{std}}(h_{\text{bck}}) \leq \mathbb{E}\left[e^{-h_b(\boldsymbol{x})h_w(\boldsymbol{x})}\right]. \tag{10}$$

Theorem 1 is proved. $\qquad\square$

### D.2 Adversarial Error Bound

We first prove Lemma 2 which is used to prove Theorem 3.

**Lemma 2.** *For any mapping $h : \mathcal{X} \mapsto \mathbb{R}$, we have*

$$\mathbb{E}\left[\max_{\boldsymbol{x}' \in B_p(\boldsymbol{x},\varepsilon)} e^{-yh(\boldsymbol{x}')}\right] \leq \mathbb{E}\left[\max_{\boldsymbol{x}' \in B_p(\boldsymbol{x},\varepsilon)} e^{-yh(\boldsymbol{x})}e^{-h(\boldsymbol{x})h(\boldsymbol{x}')}\right], \tag{11}$$

*where $\boldsymbol{x} \in \mathcal{X}$ is the input, $y \in \{-1, +1\}$ is the corresponding label, and $\varepsilon$ is the bound of allowed adversarial perturbation.*

*Proof.* Lemma 2 aims to describe the inherent feature of a hypothesis on adversarial tasks. It bounds the adversarial error of a hypothesis with its standard error and its disagreement between standard and adversarial examples. Formally, it can be written as

$$\mathbb{E}\left[\max_{\boldsymbol{x}' \in B_p(\boldsymbol{x},\varepsilon)} \mathbf{1}\left(yh(\boldsymbol{x}') > 0\right)\right] = \mathbb{E}\left[\mathbf{1}\left(yh(\boldsymbol{x}) > 0\right)\right] + \mathbb{E}\left[\max_{\boldsymbol{x}' \in B_p(\boldsymbol{x},\varepsilon)} \mathbf{1}\left(yh(\boldsymbol{x}) > 0,\ h(\boldsymbol{x})h(\boldsymbol{x}') \leq 0\right)\right]. \tag{12}$$

By applying Eq. 5 again, we have

$$\mathbb{E}\left[\max_{\boldsymbol{x}' \in B_p(\boldsymbol{x},\varepsilon)} e^{-yh(\boldsymbol{x}')}\right] \tag{13}$$
$$\leq \mathbb{E}\left[e^{-yh(\boldsymbol{x})}\right] + \mathbb{E}\left[\max_{\boldsymbol{x}' \in B_p(\boldsymbol{x},\varepsilon)} e^{-h(\boldsymbol{x})h(\boldsymbol{x}')}\right] \tag{14}$$
$$= \mathbb{E}\left[\max_{\boldsymbol{x}' \in B_p(\boldsymbol{x},\varepsilon)} e^{-yh(\boldsymbol{x})}e^{-h(\boldsymbol{x})h(\boldsymbol{x}')}\right]. \tag{15}$$

Lemma 2 is proved. $\qquad\square$

**Theorem 3.** *Let $h_{\text{bck}}(\boldsymbol{x}) = h_b(\boldsymbol{x})$ be a standard hypothesis, $h_{\text{hyb}}(\boldsymbol{x}) = h_b(\boldsymbol{x}) + h_d(\boldsymbol{x})$ be a dilated hypothesis, $\mathcal{R}_{\text{std}}(h_{\text{bck}})$ be the standard error of $h_{\text{bck}}$, and $\mathcal{R}_{\text{adv}}(h_{\text{hyb}})$ be the adversarial error of*

$h_{\texttt{hyb}}$. *For any mapping $h_b, h_d : \mathcal{X} \mapsto \mathbb{R}$, we have*

$$\mathcal{R}_{\texttt{adv}}(h_{\texttt{hyb}}) \leq \mathcal{R}_{\texttt{std}}(h_{\texttt{bck}}) + \mathbb{E}\left[ \max_{\boldsymbol{x}' B_p(\boldsymbol{x},\varepsilon)} e^{-y h_b(\boldsymbol{x})} \left( e^{-h_b(\boldsymbol{x}) h_b(\boldsymbol{x}')} e^{-y h_d(\boldsymbol{x}')} - 1 \right) \right]. \tag{16}$$

*where $\boldsymbol{x} \in \mathcal{X}$ is the input, $y \in \{-1, +1\}$ is the corresponding label, and $\varepsilon$ is the bound of allowed adversarial perturbation.*

*Proof.* In Theorem 3, we directly compare the adversarial error $\mathcal{R}_{\texttt{adv}}$ of $h_{\texttt{hyb}}$ and the standard error $\mathcal{R}_{\texttt{std}}$ of $h_{\texttt{bck}}$. Formally, it can be written as

$$\mathcal{R}_{\texttt{adv}}(h_{\texttt{hyb}}) - \mathcal{R}_{\texttt{std}}(h_{\texttt{bck}}) \tag{17}$$

$$= \mathbb{E}\left[ \mathbf{1}(\exists \boldsymbol{x}' \in B_p(\boldsymbol{x},\varepsilon), \textbf{ s.t. } y h_{\texttt{hyb}}(\boldsymbol{x}') \leq 0) \right] - \mathbb{E}\left[ \mathbf{1}(y h_{\texttt{bck}}(\boldsymbol{x}) \leq 0) \right] \tag{18}$$

$$\leq \mathbb{E}\left[ \max_{\boldsymbol{x}' B_p(\boldsymbol{x},\varepsilon)} e^{-y h_{\texttt{hyb}}(\boldsymbol{x}')} \right] - \mathbb{E}\left[ e^{-y h_{\texttt{bck}}(\boldsymbol{x})} \right] \tag{19}$$

$$= \mathbb{E}\left[ \max_{\boldsymbol{x}' B_p(\boldsymbol{x},\varepsilon)} e^{-y h_{\texttt{hyb}}(\boldsymbol{x}')} - e^{-y h_{\texttt{bck}}(\boldsymbol{x})} \right] \tag{20}$$

By applying Lemma 2, we have

$$\mathcal{R}_{\texttt{adv}}(h_{\texttt{hyb}}) - \mathcal{R}_{\texttt{std}}(h_{\texttt{bck}}) \leq \mathbb{E}\left[ \max_{\boldsymbol{x}' B_p(\boldsymbol{x},\varepsilon)} e^{-y h_{\texttt{b}}(\boldsymbol{x})} \left( e^{-h_{\texttt{b}}(\boldsymbol{x}) h_{\texttt{b}}(\boldsymbol{x}')} e^{-y h_{\texttt{w}}(\boldsymbol{x}')} - 1 \right) \right]. \tag{21}$$

Theorem 3 is proved. □