# OpenReview forum: "Neural Architecture Dilation for Adversarial Robustness"
_NeurIPS.cc/2021/Conference — NeurIPS 2021 Poster_

### Official Review · Reviewer_babp · 2021-07-11

**Rating:** 6
**Confidence:** 4

**Summary:**

This paper proposes a method termed Neural Architecture Dilation for Adversarial Robustness  (NADAR) which adversarially searches for a dilation architecture for the backbone network. Beside the dilation, a standard performance constraint is also introduced to protect the standard accuracy on clear samples form dropping too much, and a FLOPs-awareness mechanism is utilized to lower computational overhead. Theoretical analyses and emperical results validate that NADAR can improve adversarial robustness at almost no cost of generalization.

**Ethical Concerns:**

No ethical issues.

**Limitations And Societal Impact:**

Limitations: Please carefully refer to Main Review.

Potential negative societal impact: This paper does not discuss the potential negative societal impact. However, I think the proposed method raises computational cost and requires larger storage sources, thus posing potential risks to environmental protection.


**Main Review:**

Pros:
1.	The paper is well organized. Studying adversarial robustness from the perspective of neural architecture dilation is novel to solve the well-known accuracy-robustness trade-off, and is orthogonal to the adversarial training methods studied from the view of objective and optimizer.
2.	Comprehensive experimental results show that the dilation can effectively improve adversarial robustness, and the generalization performance constraint can guarantee a competitive standard accuracy. Those experimental results can support their main claims.

Cons:
1.	In Table 1, please explain the meaning of “Arch.” and “Back.”. It is better to clearly show the percentage of capacity has been increased by NADAR compared to the backbone network.
2.	It is better to report robust accuracy against stronger attacks, e.g., CW attacks[1] and AutoAttack, in Table 1-3.
3.	The paper only discusses how to apply NADAR on ResNet-like backbones. It seems to be a limitation of this paper. Please provide some discussions about utilizing the proposed method based on the backbones those do not fit the ResNet-like architecture.

[1] Towards Evaluating the Robustness of Neural Networks. SP 2017

**Time Spent Reviewing:**

3

---

> ### Author Response · Authors · 2021-08-09
> **Response to Reviewer #babp**
>
> **1. The percentage of capacity increased**
>
> The “Back.” and “Arch.” means the parameter number or FLOPs of the backbone architecture (WRN32-10) and the searched architecture, respectively. This demonstrates the amount of capacity increased by NADAR. NADAR-A increase the parameter number and FLOPs by 7.79% and 8.96%, respectively. NADAR-B increase the parameter number and FLOPs by 9.52% and 10.45%, respectively.
>
> **2. Stronger attacks**
>
> We report the AutoAttack results in a separate table, because the results of NAS methods are not available. However, we report the AutoAttack results of adversarial training methods from Table 1 in our Table 4, including PGD-7, FreeAT-8 and TRADES-6. We do not include CW and CW+, because PGD is proved to be a stronger stack than it [1].
>
> **3. Other backbones**
>
> Because Wide ResNet is the most commonly used architecture in adversarial training [1, 2, 3], we follow the previous works and use it. Other architectures are rarely included in previous works, so we also do not consider them. However, we test backbones beyond the Wide ResNet, including other variants from the ResNet family in Table 2 of our supplementary material.
>
> **Reference**
>
> [1] Towards Deep Learning Models Resistant to Adversarial Attacks, ICLR 2018.
>
> [2] Theoretically principled trade-off between robustness and accuracy, ICML 2019.
>
> [3] Adversarial training for free!, NIPS 2019.

---

### Official Review · Reviewer_DKEh · 2021-07-15

**Rating:** 4
**Confidence:** 5

**Summary:**

The paper proposes a neural architecture search that learns a residual (called dilation in the paper) to an existing network in order to improve the robustness to adversarial attacks. The loss of the search is based on improving the robustness to the attack and retaining the original network accuracy. They show improvement for wideresnet for various datasets.

**Ethical Concerns:**

nothing

**Limitations And Societal Impact:**

not related

**Main Review:**

The idea of using NAS for improving robustness is very interesting. Yet, as the authors mention, it was already done before.
While this does not eliminate the contribution of the paper, there are several problems in the current form of the paper that should be addressed before it fits for publication:

1. Theory: The authors presents some results that are supposed to support the proposed approach. Yet, these results are very straight forward to achieve so they are not really novel. But even if they are novel, they don't really analyze the proposed approach. First, They give only an upper bound to an upper bound of the loss. Second, they analyze a different architecture than the one used, which is basically a summation of two networks, which is not the case in this work. So, therefore, the theory part is not really relevant or explains the success. So right now it does not really support the paper results but just present some inequalities that I am not sure if they are helpful to something else or not. This should be shown.

2. The empirical results are limited due to the following two reasons:
a. In any architecture search there should be comparison to the best random architecture selected from the search space. This comparison is missing.
b. There is no comparison to adversarial training of other searched architectures. The table in the paper just show their results without adversarial training. This is not a fair comparison.
c. Is the improvement just because of  the search or because of the adversarial training? Did the authors tried to use TRADES also for the other NAS based approaches for adversarial training?
d. Why not all results are reported for DARTS?

3. Ablation should be done for searching for the whole architecture with the current loss with the same number of parameters?

4. What if you use a larger resnet that has the same number of params like the resnet+search and train it with DARTS?

5. Finally, in the paper Intriguing properties of adversarial training at scale (ICLR 2020) it is shown that using larger networks improve adversarial training. So basically, is not this just explain the reason why the adversarial training works better here as the network is bigger? All the experiments I suggested above should be done to prove that the success achieved is not merely due to using a larger network but that the specific used search is really the source of improvement





**Time Spent Reviewing:**

2

---

> ### Author Response · Authors · 2021-08-09
> **Response to Reviewer #DKEh**
>
> **1. Theory analysis**
>
> We analyze the upper bound of the loss to find out the factors that impact standard and robust performance of neural networks, instead of calculating the exact value of the improvement brought by NADAR. For this purpose, our analysis of upper bound is meaningful and success.
>
> The same principle applies to the architecture in our theoretical analysis. Although we analyze a simplified special case of neural networks, it helps us to find the points to focus on when dilating robust neural networks. Our motivation and algorithm design are based on this analysis. To be specific, Theorem 3 motivate our idea of dilation for robustness, and Theorem 1 motivate the design of standard performance constraint to minimize the standard accuracy drops. They are well discussed in our two remarks in Section 4.
>
> Besides, it is a common practice to analyze simplified special case in theoretical studies. Performing theoretical analysis without any simplification is infeasible. One cannot directly analyze a network as large as ResNet or Wide ResNet.
>
> **2. The empirical results**
>
> *2.a. Comparison to the best random architecture selected from the search space*
>
> We compare our method to random architectures in Figure 3 of our supplementary material. The best random architectures in the search space cannot reach the performance of our NADAR. This shows that both the enlarging of model capacity and the proposed standard performance constrained search are crucial for the results.
>
> *2.b. Comparison to adversarial training of other searched architectures*
>
> They are trained with adversarial training. Except for the standard training in the first row, baselines reported in our tables are all trained with adversarial training. In fact, if they are not, they should have zero adversarial accuracy under any attack, just like the first row. Therefore, this is a fair comparison
>
> *2.c. Whether the improvement is because of the search or the adversarial training*
>
> The improvement is because of searching for dilation architectures with NADAR. Our architectures are retrained with vanilla PGD-7, instead of TRADES. Comparing to the PGD-7 trained network, our method can reach both better standard and adversarial accuracy, when we retrain our dilated architecture with the same training setting as them. Comparing to TRADES, they aim to improve the trade-off by modifying the training method, but our trade-off ratios are still better than both the TRADES-1 and 6 settings as discussed in Section 5.2, when we only use the vanilla PGD-7 to retrain our dilated architecture.
>
> Regarding NAS methods, both the standard NAS and robust NAS methods are trained with PGD-7. Therefore, this is a fair comparison, so we do not consider training them with TRADES.
>
> *2.d. All results for DARTS*
>
> DARTS is only one of our baselines, and it is relatively weak. On CIFAR-100, we compare to other stronger NAS baselines than DARTS, including RobNet and RACL, so there is no need to compare to DARTS. Regarding Tiny-ImageNet, only recent speed-up methods for adversarial robustness consider it, so we conduct experiments to demonstrate that NADAR outperforms the backbone under different adversarial training methods.
>
> **3. Searching whole architecture with the current loss**
>
> Our loss is specially design for dilation. For example, the standard performance constraint needs to compare the backbone and dilated network. It is not proper to use our loss to search a whole architecture.
>
> **4. Train a larger ResNet with DARTS**
>
> DARTS is a method to search for neural architectures. ResNet is a family of fixed architectures. DARTS cannot be used to train ResNet.
>
> **5. Whether the success is due to large network or search**
>
> We perform ablation study in Figure 3 of our supplementary material. We compare the backbone, random dilated architectures, and architectures dilated with NADAR. The result shows that randomly enlarging the architecture cannot always guarantee a better performance than the backbone, even though the random architectures also have high probability to outperform the backbone. The architectures dilated with NADAR can always outperform both the backbone and the randomly dilated architecture. This illustrates that dilation itself has the potential, but NADAR, which utilizes searching, can provide a guarantee.

---

> > ### Comment · Reviewer_DKEh · 2021-09-01
> > **Checking things just with a weak baseline for adversarial training is not enough and theory is not convincing still**
> >
> > I first would like to thank the authors for their effort in the response.
> >
> > Yet, I am still not convinced in the answer due to the following reasons:
> >
> > Theory: The results are correct for any learned classifier. Nothing here is related to networks. You give an upper bound, which is not surprising for a sum and then say that this is a good reason that sum is a good thing. Nothing in the theorem points to really thinking that the sum is better than a single one.
> >
> > Bigger networks: I didn't see in any of the answers of the reviewers something that relates to this. EfficientNet, ResNet, WideResnet, etc. all of them have larger versions. I suspect that just using their largest versions will close the gap to your network.
> >
> > Trades: In Table 4 it is clear that TRADES-6 is better than what the current trained network. i don't see a reason why the comparison to other search methods and other networks should be with a weak adversarial training technique. Right now, all the "convincing results" rely on a very weak adversarial training approach.

---

> > > ### Author Response · Authors · 2021-09-05
> > > **A new and fair comparison to baselines and more explanations regarding the theoretical analysis**
> > >
> > > # 1. Theory
> > >
> > > In any case, we would like to thank the reviewer for acknowledging the correctness and generality of our theorem. We think those “not surprising” theories are actually meaningful, and we want to address this concern from the following aspects.
> > >
> > > - Firstly, we would like to explain **what is related to networks**.
> > >
> > > 	It is a common practice to simplify neural networks into mappings or functions in the theoretical analysis [1, 2, 3]. Besides, the **major contribution** of NADAR is the idea to *dilate an existing classifier (which can be a network) for better performance*. It is possible that other types of classifiers can also benefit from such dilation. For example, dilating a decision tree with another one may also improve its robustness, but such potential won’t weaken our method. It is important that both our theories and our network share the same summation form. Both of them include a backbone, whose standard performance is good, and an additional learner is attached to the backbone for better adversarial performance. This links the theoretical analysis and the practical implementation.
> > >
> > > - Secondly, we would like to restate **our thinking** and explain **why the sum is good**.
> > >
> > > 	Our bounds are **NOT TO** say that using the summation only is a good thing but **TO** analyze how to optimize a network (or a general leaner, as mentioned above) with such a summation form to reach good performance. We will update our paper accordingly to clarify this misunderstanding.
> > >
> > > 	To be specific, we provide two upper bounds:
> > >
> > > 	- The first one in Theorem 1 reveals that **minimizing the disagreement** between the dilation architecture and the backbone on natural samples is important. It is well-known that robust models have worse standard performance than standard models [3, 4]. But what if the robust network is a hybrid network and part of it is the standard network? According to Theorem 1, if the disagreement (the last term in Eq. 26) between the standard backbone and the dilation architecture can be minimized, theoretically, the hybrid network can have a competitive performance to the standard backbone. Even though a zero disagreement is not practical, minimizing it is already effective enough, and our experimental results demonstrate that our method indeed has a smaller ratio of nature accuracy drop than the baselines by minimizing the disagreement with a **standard performance constraint**. Besides, if we compare the adversarial accuracy under similar standard accuracy (as described in the response to Reviewer #WsAL), on a setting proposed by TRADES, where the network is trained on MNIST with a small CNN, we can reach 94.33% vs. their 93.87% under PGD-40 attack, when having a similar standard accuracy (99.33% vs. our 99.34%, with $1/\lambda = 0.6$ for TRADES).
> > >
> > > 	- The second one in Theorem 3 reveals that we can **optimize the dilation architecture** with adversarial training for better adversarial performance, even though **the backbone is fixed**. The backbone is fixed because we want to maintain its good standard performance to the utmost extent. But does partly optimization as effective as optimizing the entire network? According to Theorem 3, if the dilation architecture makes larger output than the backbone on adversarial example, it can correct the overall output even the backbone makes mistakes. As the last term in Eq. 29 approaching zero, theoretically, the adversarial performance of the hybrid network can be competitive to the standard performance of the backbone. This means only optimizing the dilation part with adversarial training but maintaining the backbone fixed is as effective as optimizing the entire network with adversarial training, and this directly leads to **the form of optimization of $\boldsymbol{\theta}_d$ in Eqs. 12 and 14**. Again, even though it is not practical that the dilation architecture can fix every adversarial sample, optimizing it is already effective enough, and our experimental results in the attached table (we will explain the table later in the following response “3. TRADES”) demonstrate that our method can consistently outperform many strong baselines in terms of adversarial accuracy.
> > >
> > > - Finally, we would like to explain **why this theoretical analysis is important**.
> > >
> > > 	The analysis indeed helps us to find the factors that impact accuracy and robustness, and it inspires our designs, e.g., the adversarial optimization of the dilation part and the standard performance constraint on it. Those designs are also proved to be effective by our experimental results. The analysis is an indication that our method is not just a random lucky try but is based on careful study and analysis.
> > >
> > > # 2. Bigger networks
> > >
> > > We want to clarify that bigger networks have better performance is indeed one of our motivations and main claims. However, we also want to gently and politely point out that the reviewer might ignore a crucial aspect, **the amount of computation budget**. The increase in computational overhead brought by NADAR is marginal, while the larger versions of those networks suffer from a tremendous increase in computational cost.
> > >
> > > For example, with CIFAR-10 as the input, ResNet-34 has 1.16 G FLOPs, and if you use NADAR to dilate the architecture, the final network has 1.31 G FLOPs, which is a 12.93% increase. But if you directly use ResNet-101, it has 2.51 G FLOPs, which is about 2.16 times of ResNet-34 and 1.91 times of ResNet-34 + NADAR. As for Wide ResNet, the difference is even more obvious. WRN32-10 has 6.66 G FLOPs, and if you dilate it with NADAR, the FLOPs increase to 7.36 G FLOPs, which is a 10.51% increase (Table 1 in our paper). But if you directly use WRN70-16, the FLOPs increase to 38.76 G, which is about 5.82 times of WRN32-10 and 5.27 times of WRN32-10 + NADAR.
> > >
> > > Besides, if the computational budget is enough, you can also apply NADAR on those larger networks. For example, NADAR can increase the PGD-20 accuracy of ResNet-101 to 53.89% (Table 2 in our supplementary material) whereas the PGD-20 accuracy of the origin backbone is only 45.50%. This is a remarkable improvement (+8.39%).
> > >
> > > # 3. TRADES
> > >
> > > We want to thank the reviewer for the suggestion regarding comparison with SOTAs. There might be some misunderstandings about the “**real performance improvement**”. Therefore, we provide some new results in **Table 1** of *“New results regarding concerns about performance improvement (25 Aug.)”* (the same table along with analysis is also attached below). Besides, we also want to gently restate that it is **our networks** that are trained with weak methods instead of the baselines. The **baselines** are **SOTA methods** and are **faithfully cited** as reported in the original works.
> > >
> > > The original intention of using a weak PGD method to train our NADAR network is to emphasize the contribution of dilation. However, considering the fairness, at this time, we control the training method to be the same and compare only the architecture. To be specific, we conduct 4 groups of experiments to compare NADAR and the WRN32-10 backbone, and they are trained with PGD-7, FAT, TRADES-1, and TRADES-6, respectively. We report the natural, PGD-20, and AutoAttack accuracy. We will update the tables in our paper accordingly.
> > >
> > > # Attachment: Table 1 and analysis
> > >
> > > We report the results in **Table 1**. As can be seen, our method can consistently outperform the backbone in terms of adversarial accuracy. Regarding natural accuracy, NADAR is competitive to the backbone (only 0.52% lower on average, which is a marginal drop, given the magnitude of robustness improvement). Especially, under the TRADES-1 setting, which focuses on natural accuracy, NADAR can outperform its backbone on both natural accuracy ($89.77$% vs. $88.64$%) and adversarial accuracy ($55.13$% vs. $48.90$% under PGD-20 and $50.90$% vs. $43.01$% under AutoAttack). In the meantime, its adversarial accuracy ($55.13$%) is much closer to the one of TRADES-6 ($56.43$%), which focuses on robustness and has low ($4.85$% lower) natural accuracy, than to the one of TRADES-1 ($48.90$%), which has a similar natural accuracy.
> > >
> > > **Table 1.** *Comparison of NADAR and WRN32-10 backbone test accuracy when using various AT methods for training.*
> > >
> > > | Model           | AT Method | Natural | PGD-20         | AutoAttack    |
> > > |:----------------|:----------|:-------:|:---------------|:--------------|
> > > | Backbone        | PGD-7     | 87.25   | 45.84          | 44.04         |
> > > | NADAR (ours)    | PGD-7     | 86.23   | 53.43 (+7.59)  | 50.44 (+6.4)  |
> > > | Backbone        | FAT       | 89.34   | 46.13          | N/A           |
> > > | NADAR (ours)    | FAT       | 88.12   | 54.53 (+8.40)  | 51.37 (N/A)   |
> > > | Backbone        | TRADES-1  | 88.64   | 48.90          | 43.01         |
> > > | NADAR (ours)    | TRADES-1  | 89.77   | 55.13 (+6.23)  | 50.90 (+7.89) |
> > > | Backbone        | TRADES-6  | 84.92   | 56.43          | 53.08         |
> > > | NADAR (ours)    | TRADES-6  | 83.94   | 57.43 (+1.00)  | 55.25 (+2.17) |
> > >
> > > # Reference
> > >
> > > [1] Formal Guarantees on the Robustness of a Classifier against Adversarial Manipulation. NIPS 2017.
> > >
> > > [2] Evaluating the Robustness of Neural Networks - An Extreme Value Theory Approach. ICLR 2018.
> > >
> > > [3] Theoretically Principled Trade-Off Between Robustness and Accuracy. ICML 2019.
> > >
> > > [4] Robustness May Be at Odds with Accuracy. ICLR 2019.

---

### Official Review · Reviewer_WsAL · 2021-07-16

**Rating:** 6
**Confidence:** 5

**Summary:**

This paper focuses on improving the adversarial robustness of the backbone CNNs with good natural accuracy. Different from those
adversarial training methods, this paper employs the dilation architecture to increase the capacity of the backbone network with minimal computational overhead. This paper provides both theoretical analysis and empirical results.

**Limitations And Societal Impact:**

Yes, the authors addressed the limitations.

**Main Review:**

Originality: The authors leverage neural architecture search approach to dilate the backbone network, in order to improve adversarial robustness while maintaining natural accuracy. The dilation has interesting motivation to provide adversarial robustness with minimal computational overhead.

Quality:
1) My main concern is that the robustness of the proposed method achieved is not significantly better than prior works. I list the detailed suggestions in the Limitations setction.
     a) I suggest the authors include results under AutoAttack for all tables, since AA is stronger and can provide a more general prospective view. Besides, AA would play the biggest role after combining the four attackers, so the authors are suggested to report the standard version of AA instead of splitting it into four attacks and reporting the performance under each respectively.
     b) It seems that no improvement is achieved on top of TRADES-6. For example, in Table 1, the average between nat and pgd-20 is 70.675 while the average of NADAR-B is 69.83; in Table 2&3, it is lack of TRADES result on CIFAR-100 and Tiny-ImageNet.
     c) The authors are suggested to compare the same group of competitors on all datasets and under different scenarios to demonstrate the effectiveness of their proposed method.

2) Another major concern is the computational cost during training.  After reading the experiment part, an intuitive question comes into my mind: what is the benefit of dilating an existing graph? Should the training process be faster than standard adversarial training? The authors are suggested to provide more evidence, either on saving training cost or improving performance, to demonstrate the motivation of their proposed method.

Clarity: This paper is well organized and clearly written, which allows an easy read. The abstract and introduction directly point
out the focused research direction, the main contributions, and the overview of the paper structure.

**Time Spent Reviewing:**

2 hours

---

> ### Author Response · Authors · 2021-08-09
> **Response to Reviewer #WsAL**
>
> **1. The significancy of improvement**
>
> Firstly, our dilation can reach significantly high adversarial accuracy gain than the backbone with a marginal drop on standard accuracy, when both of them are trained with the same PGD-7 setting. Our standard accuracy is only 1.02% lower than it, when the adversarial accuracy is 4.36%, 7.59% and 7.77% higher than it under FGSM, PGD-20 and PGD-100 attacks, respectively.
>
> Then, we address each sub-question:
>
> *1.a. AA results*
>
> AutoAttack is a relatively new attack method and it does not consider NAS, so the results for NAS baselines are not available. Therefore, we report those adversarial training results in a separate table. Regarding the combining, our NADAR-B architecture can reach 50.44% AA accuracy, comparing to 44.04%, 43.21%, 41.47%, and 43.01% of PGD, FastAT, FreeAT and TRADES, respectively. We will add the results in the final version.
>
> *1.b. Improvement comparing to TRADES-6*
>
> It is unfair to compare the average accuracy only under their TRADES-6 setting. They have a parameter to control the trade-off, and it is easy to tune it for a desired average accuracy. If you consider the ratio between adversarial accuracy gain and standard accuracy drop, our trade-off ratios are better than both the TRADES-1 and 6 settings as discussed in Section 5.2. Furthermore, if we compare the adversarial accuracy under similar standard accuracy, on a setting proposed by TRADES, where the network is trained on MNIST with a small CNN, we can reach 94.33% vs. their 93.87% under PGD-40 attack, when have a similar standard accuracy (99.33% vs. our 99.34%, with $1/\lambda=0.6$ for TRADES).
>
> We consider the situation when there is an existing backbone with sufficient standard accuracy, and we aim to improve its robustness via increasing its capacity. The backbone has competitive performance on standard task, but its robustness is low. In such a scenario, the trade-off ratio is more important. Our method can reach higher robustness improvement with lower standard accuracy drop than previous methods, which proves our method is effective.
>
> TRADES do not report results on CIFAR-100 and Tiny-ImageNet, so we do not include it. Especially for Tiny-ImageNet, TRADES is very inefficient on large-scale dataset, and only recent speed-up methods consider this dataset. Therefore, we conduct experiments to demonstrate that NADAR outperforms the backbone under different adversarial training methods.
>
> *1.c. The same group of competitors*
>
> Each baseline method report results on different datasets, so we do not include the same group of competitors considering the availability and reproducibility of them. However, we manage to include the most important baselines for each experiment. For example, the CIFAR-10 is the most common dataset, so we include as much baselines as possible; on CIFAR-100, we include a standard adversarial training method (PGD-7), a speed-up method (FreeAT), and two strong Robust NAS methods (RobNet and RACL); on large-scale dataset, speed-up is important, so we compare the dilated architecture with the backbone by training with a standard standard adversarial training method (PGD-7) and two speed-up method (FreeAT and FastAT).
>
> **2. Computational cost**
>
> With WRN32-10 as the backbone and on a NVIDIA V100 GPU, searching costs around 40 GPU hours, and retraining costs around 28 GPU hours. Comparing to our baselines, PGD-7 needs 90 GPU hours, and FreeAT-8 needs 13 GPU hours. Our running time is in a reasonable range. Our training with PGD-7 is significantly faster than training a WRN32-10 backbone, because our training converges faster and requires less epochs than it.

---

> > ### Comment · Reviewer_WsAL · 2021-08-28
> > **Thank you for your response**
> >
> > Thank you for the detailed answers. The answer to Q1 is satisfactory, the answer to Q2 clarifies many things. I agree with your points and appreciate the idea of dilating the backbone network by search approach to improve robustness. I am willing to improve my score.

---

### Official Review · Reviewer_kvfV · 2021-07-17

**Rating:** 8
**Confidence:** 4

**Summary:**

This paper studied the adversarial robustness from the neural architecture perspective. In particular, the authors introduced a dilation network to increase the overall capacity of the neural network, so that it can simultaneously handle the clean examples and adversarial examples very well. A performance constraint and a FLOPs constraint are developed to guarantee the introduced dilation network can indeed benefit the accuracy and robustness with the minimal FLOPs involved. The authors have done extensive experiments to evaluate the proposed algorithm.

**Limitations And Societal Impact:**

Nothing particular

**Main Review:**

In Line 121, the authors explained that the dilation network has been searched within a NASNet-like search space. In fact, there are quite a few different search spaces in the literature. Why the authors adopt such a search space? Further, is that necessary to customize a certain search space for the problem studied in this paper?

The authors suggested that the backbone network has a fixed architecture. But why not also do some modifications to the backbone network?

I agree it is necessary to investigate the FLOPs constraint when designing the dilation network. But why this FLOPs constraint is implemented in the formate like Eq. (11). How about directly include a separated item about FLOPs in the objective function?


**Time Spent Reviewing:**

1

---

> ### Author Response · Authors · 2021-08-09
> **Response to Reviewer #kvfV**
>
> **1. Search space**
>
> We follow the most popular search space in NAS [1, 2]. This micro search space provides high flexibility to find fine-grained architectures and is proved to be promising by previous work. Besides, it can be combined with differentiable methods for efficient and gradient-based search. Therefore, we directly used it.
>
> **2. Fixed backbone**
>
> This work is based upon the precondition that we have already had an architecture with satisfactory standard accuracy. Then, we aim to study how to properly dilate the architectures for maximal robustness gain while preserving a minimal accuracy drop. The modification on backbone might have potential negative impacts on the standard performance, and the experiments show that dilation itself is effective. Therefore, we do not consider modifying the backbone.
>
> **3. FLOPs constraint**
>
> The FLOPs constraint is introduced as a weight to the adversarial loss. As we calculate the expected FLOPs of the supernet in differentiable NAS based on the architecture parameter $\alpha$, it is actually possible to optimize it as a separated item. However, that will be hard to control its magnitude. If the constraint has too high magnitude, it will have negative impact on the architecture optimization, which needs extra tuning. Therefore, the current format is the most simplified and straightforward one.
>
> **Reference**
>
> [1] Learning transferable architectures for scalable image recognition. CVPR 2018.
>
> [2] Darts: Differentiable architecture search. ICLR 2019.

---

### Author Response · Authors · 2021-08-25
**New results regarding concerns about performance improvement (25 Aug.)**

We notice there is a concern about the extent of performance improvement. Therefore, we would like to make an update on our experimental results regarding this concern.

First of all, we want to restate that in our current experiments, we use only the vanilla PGD to retrain our networks to emphasize the effectiveness of dilation. We suppose the results, especially the trade-off ratios, can support our motivation: **pursue a maximal robustness gain with a minimal accuracy drop**. However, it is still worth noting that NADAR aims to optimize the neural architecture, and the architecture is actually suitable for any adversarial training methods.

Based on the above reasons, we conduct a series of further experiments to demonstrate that our method, NADAR, can effectively improve the performance of the backbone network when using various adversarial training (AT) methods for retraining. At this time, we control the training method to be the same and change only the architecture. We suppose this would be much fairer for us. We conduct 4 groups of experiments to compare NADAR and the WRN34-10 backbone, and they are trained with PGD-7, FAT, TRADES-1 and TRADES-6, respectively.

We report the results in **Table 1**. As can be seen, our method can consistently outperform the backbone in terms of adversarial accuracy. Regarding natural accuracy, NADAR is competitive to the backbone (only 0.52% lower in average, which is a marginal drop, given the magnitude of robustness improvement). Especially, under the TRADES-1 setting, which focuses on natural accuracy, NADAR can outperform its backbone on both natural accuracy ($89.77$% vs. $88.64$%) and adversarial accuracy ($55.13$% vs. $48.90$% under PGD-20 and $50.90$% vs. $43.01$% under AutoAttack). In the meantime, its adversarial accuracy ($55.13$%) is much closer to the one of TRADES-6 ($56.43$%), which focuses on robustness and has low ($4.85$% lower) natural accuracy, than to the one of TRADES-1 ($48.90$%), which has a similar natural accuracy.

**Table 1.** *Comparison of NADAR and WRN34-10 backbone test accuracy when using various AT methods for training.*

| Model           | AT Method | Natural | PGD-20         | AutoAttack    |
|:----------------|:----------|:-------:|:---------------|:--------------|
| Backbone        | PGD-7     | 87.25   | 45.84          | 44.04         |
| NADAR (ours)    | PGD-7     | 86.23   | 53.43 (+7.59)  | 50.44 (+6.4)  |
| Backbone        | FAT       | 89.34   | 46.13          | N/A           |
| NADAR (ours)    | FAT       | 88.12   | 54.53 (+8.40)  | 51.37 (N/A)   |
| Backbone        | TRADES-1  | 88.64   | 48.90          | 43.01         |
| NADAR (ours)    | TRADES-1  | 89.77   | 55.13 (+6.23)  | 50.90 (+7.89) |
| Backbone        | TRADES-6  | 84.92   | 56.43          | 53.08         |
| NADAR (ours)    | TRADES-6  | 83.94   | 57.43 (+1.00)  | 55.25 (+2.17) |

---

### Public Comment · ~Yulong_Lin1 · 2022-11-12
**Code**

Hi there, thank you for the interesting paper. Do you have a code implementation for the paper? The code repo provided seems to be empty now: https://github.com/liyxi/nadar

---

### Decision · Program_Chairs · 2021-09-28

**Decision:**

Accept (Poster)

**Comment:**

The paper proposes a neural architecture search method to improve the adversarial robustness of the backbone networks while maintaining good natural accuracy. The key idea is to introduce a dilation architecture to increase the capacity of the backbone network with minimal computational overhead. The paper is well motivated and organized. The new solution is novel and technically sound.

However, there are several concerns on the technical details. 1) The theoretical analysis is not specifically related to the proposed method on dilation of neural networks. While analysis of a general function has some implications, neural networks have their special properties which are studied extensively in the subarea of deep learning theory. The analysis would be strengthened if the analysis could be more specific to the neural network architectures used in the proposed method. 2) The benefits of using bigger networks have been pointed out in the literature, it is not clear whether the performance improvement is due to larger networks or the search procedure or both. More analysis is needed to really show that search architecture is very important even with larger spaces, otherwise, the main contribution would not be strong enough because the idea of using larger networks is not originally proposed in this paper. 3) The proposed method is based on PGD, which is not a very recent adversarial training method. It is unclear whether the proposed method can still maintain the advantage when using more advanced adversarial training methods, especially those that have already considered the accuracy on natural data.

Overall, the paper is novel and interesting. However, given the above concerns, the novelty and significance of the paper will degenerate. Addressing the concerns needs a significant amount of work. Although we think the paper is not ready for NeurIPS in this round, we believe that the paper would be a strong one if the concerns can be well addressed.


**Consistency Experiment:**

NeurIPS has a long history of experimentation. In 2014, NeurIPS ran an experiment in which 10% of submissions were reviewed by two independent committees to quantify the randomness in the review process. This year, we repeated a variant of this experiment to see how the quality of the review process has changed over time.  This paper was part of the experiment and was therefore assigned to two committees (consisting of reviewers, an Area Chair, and a Senior Area Chair) that reached independent decisions.  If both committees made the same recommendation, this recommendation was followed. If a single committee recommended acceptance, the paper was accepted (with the exception of a few cases in which the other committee identified what we considered a fatal flaw, e.g., an error in a key result).

This copy’s committee reached the following decision: **Reject**

The other committee assigned to the paper recommended **Accept (Poster)**.  You can find the other set of reviews, along with any follow up discussion with the authors here:
https://openreview.net/forum?id=NO_cSsVghGb